# Urbanization Heat Flux Modeling Confirms It Is a Likely Cause of Significant Global Warming: Urbanization Mitigation Requirements

Alec Feinberg 

DfRSoft Research, Physics Department, Northeastern University, Boston, MA 02115, USA; dfrsoft@gmail.com

**Abstract:** Recent ground-based measurements find the magnitude of the urbanization effect on the global average annual mean surface air temperature corresponds to an urbanization contribution of 12.7%. It is important to provide modeling to help understand these results as there are conflicting concerns. This study models the global warming contribution that urbanization heat fluxes (UHF) can make due to anthropogenic heat release (AHR), and solar heating of impermeable surface areas (ISAs), with additional secondary effects. Results help explain and support ground-based observations. Climate models typically omit anthropogenic heat release (AHR) as warming estimates are below 1%. In agreement, the baseline assessment in this paper has similar findings. However, in this study, the methods of climate amplification estimates (MCAE) with data-aided physics-based amplification models are used. When the MCAE are applied at the global and microclimate levels that consider greenhouse gases (GHGs), feedback, and other secondary effects; the results show that AHR fluxes can amplify, increasing to have an estimated global warming (GW) influence of 6.5% from 1950 to 2022 yielding a 0.9% decade$^{-1}$ increase. This increasing rate due to energy consumption is found as anticipated to be reasonably correlated to the increasing population growth rate over this time. Furthermore, using the MCAEs, this paper studies heat fluxes assessment due to solar heating of unshaded impermeable surfaces including likely secondary amplification effects. Impermeable surface areas (ISAs) such as asphalt roads, roofs, and building sides have been reported with high land surface contact temperatures (LSCTs) relative to non-ISAs and significantly found to contribute to urbanization warming. Results indicate that high-temperature unshaded impermeable surfaces (including building sides) are estimated to average around 10–11 °C above the earth's ambient temperature of 14.5 °C (showing albedo ISA estimates between 0.133 and 0.115 respectively); the ISA heat fluxes with secondary effects are estimated to have about a GW influence of 6.5%. This is broken down with average contributions of 4.0% from urban ISAs and 2.5% from rural ISA heat fluxes. Asphalt road ISA heat fluxes are estimated to have about a 1.1% global warming influence. Then the total UHF effect from ISAs and AHR with secondary effects is assessed in modeling to yield a combined average GW influence of 13% helping to confirm ground-based measurement results. Several key adjustment values were used for shading, cloud coverage, and rural-to-urban ISA ratios. Microclimate GHGs and related water vapor feedback (WVF) were assessed to increase urban warming by about 50%. As well an assessment of water vapor and radiation increases from UHF is provided. This study also shows the need to incorporate urbanization heat fluxes with secondary effects into climate models and indicates the necessity for Paris Agreement urban heat flux mitigation goals. Results also found that given average climate conditions, it is possible to mitigate much of the UHI effect with an albedo increase of 0.1 that is anticipated to lower the average impermeable surface temperatures by about 9 °C. Studies show this can be accomplished with cost-effective cool roads and roofs. Although roads are only estimated to occupy 14% of ISAs, changing roads from asphalt to concrete-type surfaces would improve reflectivity by about a factor of 5 and is estimated to mitigate about 5.5% of global warming. Unfortunately, the current overuse of black asphalt on pavements and roofs is highly dangerous to our environment causing UHI increases in heatwaves, excessive temperatures, and global warming issues and should be banned. Asphalt usage also reduces opportunities for solar geoengineering of urbanization.

**Keywords:** urbanization global warming; anthropogenic heat release; UHI global warming; microclimate modeling; UHI mitigation; solar heating of impermeable surfaces

## 1. Introduction

Recent ground-based measurements by Zhang et al.'s [1] found, "The magnitude of the urbanization effect on global land averaged annual mean surface air temperature change over 1951–2018 in this study is 0.03 °C decade$^{-1}$ and the corresponding urbanization contribution is 12.7%." Part of the goal of this study is to help explain through microclimate data-aided physics-based and estimation modeling, the feasibility of their observation and provide root cause assessment. This paper is able to demonstrate modeling validation and help in our understanding of the root causes due to UHFs. The recent comprehensive work of Zhang et al. [1], the only worldwide study of its kind, has not been recognized in climate models and the IPCC AR6 report; the IPCC in AR6 [2] states that urbanization has a "negligible impact on global annual mean surface-air warming (very high confidence)." Such statements show the difficulty of understanding root causes and Zhang et al.'s [1] findings.

One may begin to understand Zhang et al.'s [1] results by realizing that population growth, energy consumption, and its $CO_2$ byproduct are highly correlated to global warming [3]. Although urbanization occupies only a small area of the earth, currently, 55% of the world's population lives in cities [4]. IEA [5] and Yang et al. [6] estimated that 75% of energy is consumed within cities. As well, automotive activity and fossil fuel usage generate increases in $CO_2$ in urban areas where Koerner and Klopatek [7] estimated that 80% of $CO_2$ originates. We also know that UHIs have high concentrations of heat fluxes that have been observed to exceed 400 Wm$^{-2}$ in city centers [8]. This increases the probability for local GHG pollution and humidity exasperation effects to interact with heat fluxes amplifying heat higher than the standard atmosphere [2,9].

In general, solar heating is the primary source of warming the earth while greenhouse gases (GHGs) are an important secondary re-radiation effect. Therefore, additional UHFs are concerning as they increase primary heating and can be amplified in the microclimate and the background global climate by secondary effects. GHGs and related water vapor feedback (WVF) and other secondary effects are important parts of climate science and need to be included in modeling.

In this study, there is also a focus on providing physics-based data-aided UHI microclimate and background global climate modeling of GHGs and related WVF and other amplification effects. This modeling can help explain how urbanization can contribute to global warming. The other key microclimate UHI effects are atmospheric aerosol pollutants, loss of evapotranspiration cooling, temperature inversion reduced cooling, and so forth.

This paper also studies anthropogenic heat release (AHR) which is understood to be a key contributor to the urban heat island. As well, this paper focuses on solar heating of ISAs such as asphalt pavements, rooftops, and the large solar area of building sides, as the other main contributor to the UHI. Therefore, in this portion of the study, assessments are limited to land cover from ISAs only which excludes areas such as farmlands. UHI heat fluxes are in the form of sensible and latent heat to the urban canopy and AHR is created mainly through heating and cooling energy consumption, running appliances, transportation, and industrial processes, which convert energy into anthropogenic heat release [10–14].

Climate models typically do not incorporate UHFs as initial assessments (without secondary effects) show only a small contribution (~2%, see for example Section 2.5) would result in global warming [2,6,14,15]. This leaves a gap in climate modeling with possible errors if these are key influences on global warming, as estimates indicate in this study and by Zhang et al. [1] measurements.

At the microclimate level, the 'energy in' for solar heating of ISA can be estimated with both the land surface contact temperatures (LSCTs) and the albedo effect. To make assess-

ments, LSCT estimates are made in this paper relative to the global ambient temperature for unshaded ISAs. High LSCT is something we all experience in walking on unshaded dark pavements. It provides a helpful additional practical metric. These heat fluxes in the UHI environment can amplify, and modeling helps in assessments. For example, Zhao et al. [16] found that cities in humid compared to dry environments are 3.3 K hotter (see Section 2.2). Their data has been modeled as UHI water vapor feedback (WVF) recently [17] and related amplification estimates are provided in Section 3.1.4.

Many land surface temperature studies and ground measurement station observations are taken from mean surface air temperatures at a height of about 5–6.5 feet [18] which are significantly lower in temperatures than LSCT from solar heating of asphalt roads, for example, that can often be higher than 20 °C above ambient. Zhang et al.'s [1] ground station measurements were taken at about this height. Given that they found a significant urbanization effect at about 5–6.5 feet, it is likely that LSCT heat fluxes from solar heating of ISAs are making substantial contributions in their observations compared to AHR.

A recent satellite study out of China by Guo et al. [19] found in 10 of 11 cities, albedo estimates appear to average about 0.16. Guo et al. [19] deduced a substantial cooling effect from the temporal history of the "relationship between land surface albedo and land cover in cities". These are land-cover types of measurements with the relationship between surface albedo and urban canopy not fully known. However, this leaves AHR and their cooling losses as the main root causes for the UHI effects in these megacities according to their results. It is interesting to note that the results in this study on heat fluxes at the global level from solar heating of ISAs, an albedo of about 0.16, found that LSCT would average about 8.5 °C above ambient for unshaded areas and impact about 2.7% of global warming.

It is important to know how comparisons are made in studies. Cities have higher microclimate local amplification compared to rural areas so equating albedo assessments is difficult. It is pointed out in this paper, that it can be helpful to incorporate amplification effects when appropriate and also consider making additional comparisons relative to ambient conditions or the natural reflectivity of land which is about 25% [20] or the earth (~30%) as this provides valuable information. Using such comparisons, for example, Guo et al. [19] would likely deduce albedo UHI warming rather than cooling effects (see Section 4.1). Further, it implies that albedo UHI goals should be higher to offset warming from solar heating of ISAs, AHR, the microclimate GHG amplification, and the cooling losses that can occur in cities.

This study also provides modeling for both AHR and ISAs which includes building effects. The impact of solar heating due to building sides and related solar canyon effects are not well understood in GW albedo land satellite studies (see Section 4.2) along with microclimate amplification effects. Often climatologists favor satellite compared to full ground-based direct measurements and these can appear in conflict with GW urbanization assessments [21] (see Section 4). In general, there is a long history of conflicts among authors on the effects of urbanization warming that modeling in this study may help resolve.

*Document Smart Diagram*

To aid the reader in this two-part study, a Table 1 smart flow chart is provided below. Each part of the study is laid out in the table.

**Table 1.** Smart Flow Chart.

| Urbanization Heat Flux Assessment | | | | |
|---|---|---|---|---|
| **Urban Flux Type** | **Amplification Models** | **Model Verifications** | **Flux Type Results** | **Results** |
| **Part 1: 3.1** **AHR GW** Baseline Data Sections 2.4, 2.5 and 3.1.9 → **Part II: 3.2** **ISA GW** Baseline Est. Section 3.2.1 → | Tables 2 and 3, Figure 1 Aerosol Etc. Section 2.3 Global background Section 3.1.1 UHI GHG: Section 3.1.3, Appendix D **Model 1** UHI WVF Sections 2.2 and 3.1.4 **Model 2** Dry & Wet Mixed Section 3.1.5 **Model 3** Footprint Sections 2.1.1 and 3.1.6 **Model 4** Rural Section 3.1.7 **Model 5** → Table 3, Figure 1 | **Wet & Dry Model 3** Footprint **Model 4** Section 3.1.10 Tables 3–5 → | **AHR 6.5% GW:** Section 3.1.8, Equation (14) → **ISA 6.5% GW:** Section 3.2.1 Table 6 → | **AHR+ISA GW** 13% GW Section 3.2.2 Table 7 |
| Mitigation of UHF Sections 3.3 and 4.1 Urban SG requirements, Section 3.3.2 Helpful Equations: Appendices A and B | **Mitigation & Helpful Information** GW Breakdown of Energy Flux, Appendix F Global background feedback, Tables 2–4 Sections 2, 2.5, 3.1.1 and 3.2.1.3 , Appendices C and F | | UHI WVF, Appendices E and F Irradiance, Section 3.2.1.1 Satellite Issues, Section 4.2, Appendix G | |

## 2. Key Data and Methods

The key data used in modeling is overviewed in Table 2 with their applicable sections.

**Table 2.** Key Data Required in Modeling.

| Data Type | Values | Description | Reference(s) | Applicable Sections |
|---|---|---|---|---|
| Fossil Fuel | 176,431 TWh | Consumption in 2021 | [22] | Section 2.4 |
| UHI WVF | 3.4 Wm$^{-2}$ K$^{-1}$ | UHI WVF in humid climates | [17] | Sections 2.2 and 3.1.4 |
| Global WVF | 1.6 Wm$^{-2}$ K$^{-1}$ | Global WVF | [23] | Section 3.1.4 |
| Global Warming | 5.1 Wm$^{-2}$, 0.95 °C | From 1950–2019 | [9] | Sections 2.5, 2.1.1 and 3.2.1.2 |
| Forcing | 2.38 Wm$^{-2}$ | From 1950–2019 | [9] | Sections 2.3, 3.1.8 and 3.2.1.2, Appendix F |
| UHI $\Delta T$ Increase | 3.3 K | UHIs $\Delta T$ increase in wet cities | [16] | Sections 2.2 and 3.1.4 |
| Cloud Percent | 47% | Irradiance through clouds | [24] | Section 3.2.1.1 |
| Unshaded% ISAs | 65%, 75% | Urban & rural unshaded %ISAs | Estimated, solar canyon | Section 3.2.1.1 |
| Global Feedback | 2.15 | WVF Amplification Factor | [9] | Sections 2.2, 2.3, 3.1.8 and 3.2.1.2 |
| Global GHG Factor | 1.62 | GHG Amplification with 62% re-radiation | [9,24] | Sections 3.1.1, 3.1.8 and 3.2.1.2 |
| UHI Footprint | 3.2 | UHI Amplification model | [25], Section 3.1.10.1 | Sections 3.1.1, 3.1.6, 3.1.10.1 and 3.2.1.2 |
| % Urban Climate | 67% vs. 33% | Cities in wet versus dry climates | [26] | Section 3.1.5 |
| ISA | 1.29 M km$^2$ | 0.255% of the earth | [27] | Section 3.1.9 |
| ISA Roads | 14% of ISA | Road % of ISA | [28] | Section 3.2.2 |
| ISA% Urban | 33%, 40%, 50% | Urban vs. Rural %ISA | [28] | Sections 3.1.9, 3.2.1.1, 3.2.1.2 and 3.2.2 |

In GHG re-radiation modeling, at the global level secondary effects can be applied to energy terms. For example, GHGs increase the warming of the earth from about −18.6 °C to 13.9 °C. In the absence of global warming, a simple re-radiation model illustrates how this amplification can be applied [9]

$$P_{Total} = P_\alpha + \overline{f} \, P_\alpha = P_\alpha \left(1 + \overline{f}\right) = A_{GHG} P_\alpha \tag{1}$$

Here $P_\alpha$ is the solar 'Energy In' equivalent to the 'Energy out' (in Wm$^{-2}$), $\overline{f}$ is the average GHG re-radiation value of about 62% [9]. In this paper, we can consider the secondary GHG re-radiation term $\left(1 + \overline{f}\right)$ as a GHG amplification factor ($A_{GHG}$), shown in Table 3 to equate to 1.62. As well, amplification due to feedback effects can be applied similarly [9]. In Table 3, an overview of secondary factors for the MCAE is presented and applied to urbanization heat flux estimates both globally and locally. The major local

secondary microclimate effect root cause from the table occurs in humid environments. Furthermore, studies suggest that local $CO_2$ GHGs can impact this UHI intensity in both dry and humid environments. In Sections 2.1–2.3 and 3.1.2–3.1.5, overviews and assessments are provided on the influence of these local urban secondary effects.

**Table 3.** Global and local urban secondary factors for the MCAE.

| Secondary Effect | Amplification Factor | Reference |
|---|---|---|
| Global Amplification Estimates | | |
| Global re-radiation GHGs | 1.62 | [9,24] |
| Global feedback (water-vapor and other effects) | (>2) 2.15 | [9,23,29,30] |
| Global combined amplification effects | 3.48 = 1.62 × 2.15 | |
| Microclimate Amplification Estimates (humid vs. dry climates) | | |
| Physics-based Modeling | 1.2 | Model 1: Section 3.1.3 dry climates |
| Physics-based Modeling | 2.55 | Model 2: Section 3.1.4 humid climates |
| Physics-based Modeling | 2.2 | Model 3: Section 3.1.5, mixed climates |
| UHI Estimate | 3.2 | Model 4 (footprint): Section 3.1.6 |
| Rural Area Estimate | 1.1 | Model 5: Section 3.1.7 |
| Global × Microclimate (Local) Amplification Estimates | | |
| Global & UHI local combined amplification effects | 1.62 × 2.15 × $A_{Local}$ | Sections 3.2.1 and 3.2.2, (Example in Equation (6)) |

As an overview of the MCAE, in this paper, the general microclimate model mixed with global background climate amplification factors is as follows

$$P_{A-UHF} = ([A_U P_{ISA-U} + A_{Ru} P_{ISA-Ru}] + [A_U P_{AHR-U} + A_{Ru} P_{AHR-Ru}])_{Micro} A_{GR} A_{GF} \quad (2)$$

In the equation, $P_{A-UHF}$ is the total amplified urban heat flux. In the microclimate portion right-hand side of the equation $A_U$, $A_{Ru}$ are the urban and rural amplification factors, $P_{ISA-U}$, $P_{ISA-Ru}$ are the urban and rural solar heated ISA fluxes, $P_{AHR-U}$, $P_{AHR-Ru}$ are the urban and rural AHR fluxes. The microclimate heat fluxes are released into the atmosphere as out-going longwave radiation and then can be amplified by the global background climate assessed by $A_{GR}$, and $A_{GF}$, the global background re-radiation and feedback amplification factors of 1.62 and 2.15, respectively in Table 3. Table 3, $A_U$ is assessed by Model 3 and 4 separately and averaged while $A_{Ru}$ is identified as Model 5. Stored sensible heat emissions from buildings with large heat capacities are typically released over the daily cycle (Section 3.2.1.1) and therefore intrinsically included in the $P_{ISA}$ and $P_{AHR}$ terms.

UHFs from ISAs are dominated by longwave radiation and sensible heat to the atmosphere while AHR has been broken down by some authors with a small portion resulting as latent heat. However, in climate modeling, it is often treated as consisting of longwave radiation and/or sensible heat [13,31] to the atmosphere. We assume the heat absorbed in the lower atmosphere will turn up into longwave radiation and treat it as other authors do in this type of climate modeling [32] as longwave radiation.

*2.1. The Global Background, Microclimate, and the Global Warming Flux*

2.1.1. Global Background Amplification

All UHFs are subject to increase due to the global background climate. There are two global climate amplifying effects. UHFs can be increased by the re-radiation of GHGs. About 62% re-radiation is anticipated yielding a 1.62-factor increase. UHFs can also be

increased in the global climate by feedback that is dominated by water vapor. This is a 2.15 factor. Both of these effects are discussed in Section 3.1.1 and explicitly shown in each calculation where applicable. When the background global climate is included in UHF estimates, global warming percentages due to urbanization effects are then estimated relative to 5.1 $Wm^{-2}$ which represents the radiative warming due to forcing and feedback that occurred from 1950 to 2019. This number is broken down in Appendix F and Equations (5)–(7) illustrate its use.

### 2.1.2. Estimating Microclimate UHI Amplification

Physics-based UHI microclimate modeling and estimations (See Figure 1) supported by data are provided in Section 3.1.3, Section 3.1.4, Sections 3.1.5–3.1.7. There are five models: Model 1 for dry climates, Model 2 for humid areas, Model 3 for mixed-weighted dry-humid climates, Model 4 is a footprint amplification type estimate added to Model 3, and Model 5 for rural GHG dry and wet microclimates.

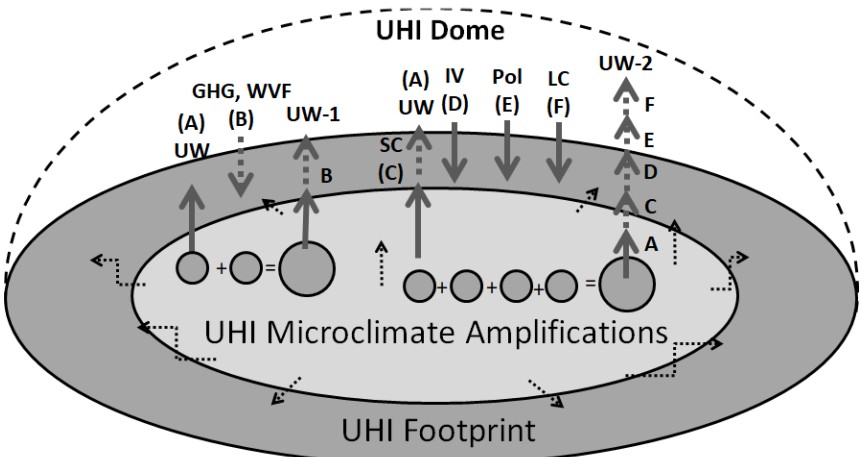

**Figure 1.** Visualization of amplifications of main heat fluxes in UHIs (not to scale). Arrows A is the main upwelling (UW) from initial heat flux either due to ISA solar absorption and/or AHR. B is the downwelling from secondary WVF with GHGs, $CO_2$, and/or methane. Then UW-1 is the resulting total amplified heat fluxes from A and added B due to GHG re-radiation. On the right portion, the dashed line A, has added upwelling C component resulting from the solar canyon absorbed shortwave radiation. The other types of secondary downwelling or intensifying heat are D, due to possible temperature inversions (IV), E pollution (Pol), and F loss of cooling (LC) effects such as wind reduction, loss of evapotranspiration, etc. Then UW-2 is the resulting total amplified heat fluxes from main fluxes A and added secondary effects from C, D, E, and F. The resulting combined amplification magnitude (small dashed arrows) is often assessed with UHI footprint and dome measurements. The inner circle is the city limits (light gray), and the outer circle (dark gray) is the observed footprint from heat spreading beyond the city limits.

It is assumed in this study that UHIs dominate urban effects. The initial work on microclimate amplification by the author [25], assessed UHI amplification factors ($A_{UHI}$) based on UHI footprint [33] and dome estimates [34]. These are depicted in Figure 1. In this paper, modeling with data is used to help justify the prior footprint amplification value of 3.2 (Model 4) with the data assessment in Section 3.1.10.1. This value is also found to confirm Zhang et al.'s [1] findings yielding modeling consistency.

Zhou et al. [33] found an average footprint of 3.2 times the urban area. Feinberg [25] estimated that the overall urban areas growth from 1950 to 2019 increased also by a factor of 3.2 and dome areas increase by a factor of 8.4. The footprint and dome growth are indications of amplified heat flux that is observed to spread beyond the boundaries of the city itself both horizontally and vertically, respectively as depicted in Figure 1. As Figure 1 shows, heat fluxes amplify and increase in UHI microclimate for several reasons

including local water vapor feedback (WVF) or GHG re-radiation, solar canyon effect, heat inversions, pollution, and loss of cooling. These amplify heat spreading beyond the area of the city itself creating UHI footprints and dome effects. Zhou et al. [33] warn that "ignoring the footprint (FP) may underestimate the UHI intensity in most cases. Yang et al. [6] in a temporal study from 2003 to 2016 on 302 cities in China showed that day-night footprints are rapidly growing at an alarming rate of about 4.4% per year. This is an indication of a rapid spreading of 'heat pollution' (i.e., excess heat fluxes) in China and may also be occurring worldwide. It is an indication of the increasing local amplification issues as well. In this study, we provide a mixture of data-aided GHG physics-based amplification models and also use additional estimates for other effects which help support Zhang et al. [1] study.

*2.2. UHI Amplification in Humid Environments*

Many authors have reported more intense and frequent heatwave (HW) issues in humid compared to dry UHIs [35–43]. Authors found a strong amplification effect, for example, Russo et al. [38] found "the magnitude and apparent temperature peak of heatwaves, such as the ones observed in Chicago in 1995 and China in 2003, have been strongly amplified by humidity". In the very humid wet climate of Bangladesh, Dewan et al. [44] found, "Results indicated that annual surface UHI intensity was greater in the larger cities of Dhaka and Chittagong than in the smaller cities. Surface UHI intensity observed during the day was also greater than at night." Wang et al. [34] studied intensified humid heat events under global warming and found "Humid heat events show intensifications to dry heat events with higher frequency, duration, and intensity".

Empirical HW evidence exists for synergistic effects [37,40]. Liao et al. [40] in a study on HW exacerbation from UHIs in wet versus dry climates in China also showed a strong synergistic heatwave effect observed both in intensity, duration, and frequency. Liao [40] also noted that "in wet climates, the increasing trends of HWs in urban areas are greater than those in rural areas, suggesting a positive contribution of urbanization to HW trends".

Li and Zeid [37] studied the Baltimore-Washington metropolitan humid areas and found "synergies between urban heat islands and heatwaves. That is, not only do HWs increase the ambient temperatures, but they also intensify the difference between urban and rural temperatures. As a result, the added heat stress in cities will be even higher than the sum of the background urban heat island and the heatwave effect".

Zhao et al. [16] compared similarly constructed cities with twenty-four located in the humid southeastern United States to 15 cities in dry climates. They found an average $\Delta T$ increase of 3.3 K observed in daytime hours in humid climates with little differences in nighttime hours. This data is key in supporting Models 2, 3, and 4. Zhao et al. [16] attributed the humidity amplification effect to a reduction in sensible heat convection cooling efficiency. However, the observations by Zhao et al. [16] and other authors such as Dewan et al. [44] found the humidity effect occurs primarily in the daytime suggesting other mechanisms as well related to an atmospheric UHI humidity amplification mechanism. In a recent atmospheric study by Zhang et al. [45], urbanization atmospheric effects were observed to have a strong complex influence on extreme surface air temperature, where parameters studied were cloud cover, water vapor pressure, and sunlight duration. In their study, water vapor pressure was a key factor for humidity warming issues. Yang et al. [14] in east China (a humid environment in summer) found, "Overall, urbanization contributes to more than one-third of the increase of intensity of extreme heat events in the region, which is comparable to the contribution of greenhouse gases".

We suggested in prior work [17] and this paper that the UHI humidity effect is also likely a dual mechanism with GHG re-radiation and this can also relate to UHI convection cooling efficiency losses. Indeed, in daytime hours, solar heating fluxes are strongly increased in UHIs, and since warm air holds more water vapor in humid environments (see Section 3.3); this suggests that the $\Delta T$ increase (~3.3 K) in cities relative to rural environments in humid vs. dry climates, supports a likely GHG mechanism as well.



Independent of the actual mechanism, the water vapor feedback equation is a function of the specific humidity and temperature changes. To quantify Zhao et al. [16] data, Feinberg [17] assessed it with a UHI temperature difference water vapor feedback model. The mathematical treatment found a UHI local feedback value of 3.4 $Wm^{-2}\,K^{-1}$ [17] for cities in humid environments at 15 °C. This is about a factor of 2.1 higher than standard atmospheric global estimates (see Section 3.1.4, Equation (8)) but is much lower than the average UHI footprint and dome-expanded thermal area estimates discussed in Section 2.1. The temperature difference feedback model was verified by using it to reproduce global water vapor feedback estimates and is discussed in Section 3.1.4 with data verification in Section 3.1.10.

### 2.3. Urban Pollution of Aerosol, Haze, and other Similar Effects

Aerosols concentrations can reflect, and absorb solar radiation, and can reduce the amount of shortwave radiation reaching the ground producing a UHI cooling effect [46–48]. Guo et al. [19] observed a cooling effect in megacities in China. Aerosol reflectivity may have contributed to albedo increases.

However, Cao et al. [49] found that "the biogeochemical effect of urban aerosol or haze pollution is also a contributor to the UHI depending on particle size. Our results are based on satellite observations and urban climate model calculations. We find that a significant factor controlling the nighttime surface UHI across China is the urban-rural difference in the haze pollution level. The average haze contribution to the nighttime surface UHI is $0.7 \pm 0.3$ K (mean $\pm$ 1 s.e.) for semi-arid cities, which is stronger than that in the humid climate due to a stronger longwave radiative forcing of coarser aerosols".

Aerosols and haze pollution effects differ from city to city and are not easily modeled. These are perhaps best assessed as part of footprint and dome observations shown in Figure 1 (see Section 2.1 and Model 4).

### 2.4. AHR Data and Method

The total global energy production and consumption increase for coal, oil, natural gas, and other sources were calculated by Ritchie and Roser [22] up to the year 2021; this is shown in Figure 2 by the solid line. In this paper, it is assumed that 90% of this energy consumption ends up as heat (discussed below). A focus is on the time period relative to 1950 since this is suggested by the knee of the curves in Figure 2 and about the period of time covered in Zhang et al.'s [1] study. The annual AHR portion in 2021 is simply divided by the total area of the earth to obtain the global warming power per unit area and then compared to the value for global warming from 1950 to 2021. Then we considered both local and global secondary amplification effects in Table 3 based on prior studies [50] and modeling in Section 3.

We note in Figure 2, there appears a reasonable correlation between energy consumption shown by the solid line [22], and the global warming dashed line [51]. This correlation warrants further investigation [3]. It is also interesting that the slope of the energy curve is about 2196 TWh/year from 1975 to 2021, and in the last 5 to 10 years, the yearly amount has averaged 2133 TWh/year indicating the high rate of energy consumption increase per year has not significantly slowed.

Many authors approximate that 100% of the energy consumed eventually turns into heat. For example, Flanner [15] states, "Most energy used for the human economy is immediately dissipated as heat, either because of inefficient conversion processes or to serve the purpose of heating. Utilizing the second law of thermodynamics, it is assumed that all non-renewable primary energy consumption is dissipated thermally in Earth's atmosphere". According to the second law, there are no reversible processes so heat dissipation always occurs. According to the first law, energy use relates to some kind of loss of internal energy at the system's source supplying the energy which can go into work, heat, and sometimes part of the energy may be stored elsewhere. However, the work conducted relates to some type of kinetic energy change which eventually dissipates through frictional

losses yielding more heat. We note that while many manufacturing processes dissipate heat, useful energy also creates long-lasting products such as buildings (with mgh potential energy), batteries, etc., which have some new stored internal energy. Other energy escapes to outer space without causing any heating similar to city night light energy. However, the actual energy consumed that turns readily into heat is not well established but is always estimated to be a high percentage (usually 100%) in climate models [14,15,52]. According to Ritchie and Roser [22], for example, a large portion of the energy consumed is in the northern areas, likely used primarily for direct heating. To be reasonably consistent with other authors and assess feasibility, we assume 90% of consumed energy eventually turns into heat.

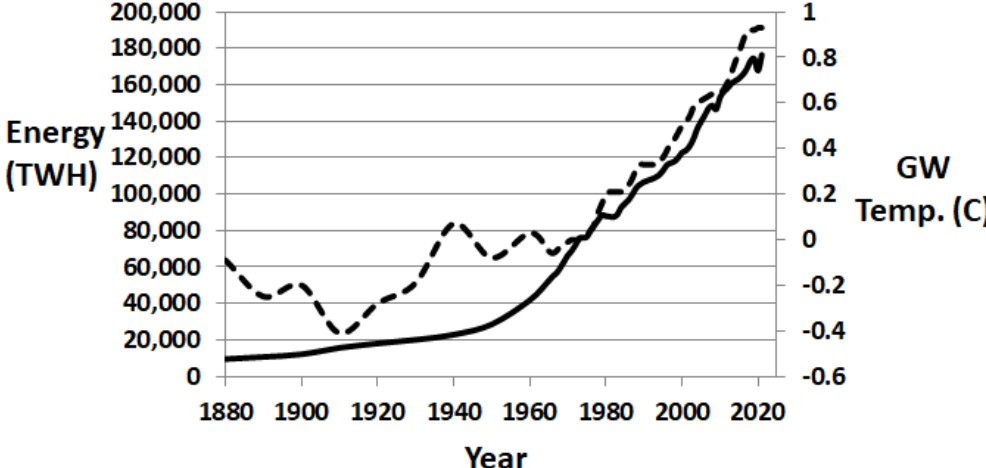

**Figure 2.** Global energy consumption [22] in TWh (**left** axis solid line) compared to global warming trend [51] in °C (**right** axis dashed line).

*2.5. AHR Baseline Estimates and Global Background Amplification*

The total global energy consumption calculated by Ritchie and Roser [22] by 2021 has increased to 176,431 TWh annually (see Figure 2). Using a factor of 90% for an estimate of energy consumption converted to AHR, we find that annual power per year dissipated is

$$p_{AHR}(2021) = (176,431 \times 10^{12} \text{ Wh}/8760 \text{ Hrs}) \times 90\% = 18.13 \times 10^{12} \text{ Watts} \qquad (3)$$

Here small $p_{AHR}$ denotes the AHR power dissipated. Considering the area of the earth, then the total global warming (GW) energy per unit area denoted by capital $P_{AHR}$ is

$$P_{AHR}(2021) = 18.13 \times 10^{12} \text{ Watts}/510 \times 10^{12} \text{ m}^2 = 0.0356 \text{ Wm}^{-2} \qquad (4)$$

By 2021, the estimated GW since 1950 is about a 0.95 °C temperature rise which approximately corresponds to the time period in the Zhang et al. [1] study. Then the 0.95 °C temperature rise equates to about 5.1 Wm$^{-2}$ (see for example Feinberg, [9], or one can use the Stefan-Boltzmann Law). This yields a baseline percentage of GW due to AHR yielding

$$\%P_{AHR}(2021) = 0.036 \text{ Wm}^{-2}/5.1 \text{ Wm}^{-2} = 0.71\% \qquad (5)$$

Therefore, without considering any secondary effects, baseline data indicates about a 0.7% global warming influence which is comparable to what other authors have reported [14,15].

## 3. Results

We divide calculations into two sections for contributions from AHR (Part 1) and solar heating of ISAs (Part 2) for global warming estimates (see the smart diagram in the introduction). We start with AHR assessments as the calculations are simpler. Data-

aided physics models and estimation assessments of microclimate amplification effects are also found in Part 1 (Sections 3.1.2–3.1.7). As well in Appendix F, we discuss UHF radiation forcing with water vapor increase estimates. In Section 3.3, UHF mitigation goals are presented.

*3.1. Part 1: AHR Estimates*

3.1.1. AHR Global Secondary Effects

Global background climate secondary estimates for greenhouse gases and feedback have been assessed in climate science IPCC models [24] and by the author [9]. Then, it is simplest to use well-established traditional climate science background global secondary estimates first. As indicated in Table 3, the potential global background average greenhouse gas re-radiation is 62% yielding a heat fluxes amplification factor of 1.62 in Equation (1) [9,24]. As well feedback (primarily due to water vapor) has the potential to more than double the heat flux forcing [23,29,30]. A 2.15 feedback amplification factor has been estimated by Feinberg [9]. These global secondary effects can increase AHR flux in Equation (4) warming to

$$P_{AHF-GS}(2021) = 0.036 \text{ Wm}^{-2} \times 1.62 \times 2.15 = 0.124 \text{ Wm}^{-2} \tag{6}$$

Here $P_{AHR-GS}$ is the AHR with the background global secondary effects (in which we have relatively high confidence in global amplification estimates in climate science).

Similar to Equation (5), we can divide Equation (6) by 5.1 Wm$^{-2}$ to obtain the percentage of GW due to AHR with global secondary effects to find

$$\%P_{AHF-GS}(2021) = 0.124 \text{ Wm}^{-2}/5.1 \text{ Wm}^{-2} = 2.43\% \tag{7}$$

This yields the percentage of GW due to AHR when the global background climate amplification is applied.

Note the 5.1 Wm$^{-2}$ consists of forcing and feedback that is detailed in Appendix F for clarity. Therefore, urban estimates in this paper, as shown in Equation (6), are amplified by the global background re-radiation as well as the feedback factor 2.15. That is, feedback approximately doubles the urbanization forcing (similar to $CO_2$ energy fluxes). Therefore, as shown in Equation (6) urban forcing percentages are increased by the re-radiation factor 1.62 and the 2.15 global feedback effect. This means that percentages are relative to the 5.1 Wm$^{-2}$ as in Equation (7) and not relative to total forcing estimates which can be uncertain. In all calculations, this is explicitly shown including in the Tables (where applicable).

3.1.2. UHI Microclimate Amplification Models

Cities amplify heat. In general, any hotspot on the earth, such as Death Valley or UHIs may amplify or intensify solar heat. For example, a common reason that land intensifies heat is often due to loss of cooling (LC) from wind reduction or evapotranspiration loss (Figure 1F). This is common in UHIs. A key microclimate amplification mechanism considered in this study is related to UHI GHGs (Figure 1B) occurring in dry (Section 3.1.3) and humid climates (Sections 2.2 and 3.1.4). Other UHI amplification mechanisms relate to temperature inversions, haze/aerosol pollution, and loss of cooling (Figure 1D–F).

There are five models provided to assess microclimate amplification effects in Sections 3.1.3–3.1.7. Models are mainly data justified in these sections and Section 3.1.10.

3.1.3. UHI GHG Amplification Estimate in Dry Climates: Model 1

From Equations (1) and (4), we note that due to the high average AHR energy in UHIs (see Equation (14)), and increases in GHGs (mainly $CO_2$ in dry climates, see Appendix D) compared to rural areas, there is a higher probability for IR absorption by $CO_2$ molecules leading to increases in subsequent re-radiation events compared to the standard atmosphere. We note that due to its density, $CO_2$ is increased in the lower atmosphere. As well, automotive activity and fossil fuel usage generate increases in $CO_2$ in urban areas

where 80% of $CO_2$ is estimated to occur as discussed in Appendix D. This can amplify and increase heat fluxes higher than the standard atmosphere. Based on an overview of increased $CO_2$ data observations in cities in Appendix D, we take this increased GHG concentration (mainly due to $CO_2$) to average conservatively about 20%. This value is justified in Section 3.1.10 for surface temperature change due to $CO_2$ observations related to Table 4. In that section, data is cited on over 300 megacities in China during the lockdown by Liu et al. [23] when $CO_2$ was minimal reducing temperatures. Their data matched fairly well with the proposed Model 1. Here in Model 1 data in Appendix D leads to a 1.2 factor in GHG concentration increase in cities compared to the standard atmosphere. Recall, that exact estimates are not part of the goal, providing feasibility and root cause estimates aided by data are the main objectives of this study. Then a simplistic physics-based local amplification estimate can be made for cities relative to the standard background atmosphere, where in the microclimate we estimate an increase in atmospheric re-radiation amplification of

$$A_{U-Dry} = \frac{\overline{C}_{GHG-U}}{\overline{C}_{GHG-S}} = 1.2 \qquad \textbf{Model 1} \qquad (8)$$

Here $\overline{C}_{GHG-U}/\overline{C}_{GHG-S}$ is simply the ratio of the estimated increase of GHGs (primarily $CO_2$) in urban compared to the standard atmosphere taken as 1.2 and as stated, this value is justified in Section 3.1.10 by $CO_2$ data observations (Appendix D) with resulting temperature changes compared well to model predictions in Table 4. Model 1 can be used both in dry and humid atmospheres.

3.1.4. UHI Amplification Estimate in Humid Climates: Model 2

As discussed in Section 2.2, UHI temperatures are increased in humid versus dry climates relative to rural areas. At the background global atmospheric level, there are several estimates from different authors on the water vapor feedback (WVF) effect. For example, Gordon et al. [30] estimated 2.4 $\text{Wm}^{-2}\,\text{K}^{-1}$, Dessler, et al. [29] found 2.2 $\text{Wm}^{-2}\,\text{K}^{-1}$, and Liu et al. [23] estimated a value of 1.6 $\text{Wm}^{-2}\,\text{K}^{-1}$. In Section 2.2, a discussion assessed that microclimate UHI amplification was perhaps best quantified mathematically with a recently provided temperature difference water vapor feedback model by the author [17] that could be applied to the mean $\Delta T$ observations from Zhao et al., [16] data in humid compared to dry UHI climates. When the WVF model was applied to Zhao et al. [16] temperature difference data for cities in humid environments that averaged 3.3 K hotter than dry environments, results found that in humid cities, the local water vapor feedback at 15 °C averaged 3.4 $\text{Wm}^{-2}\,\text{K}^{-1}$ (Feinberg, 2022b). This is perhaps a best method for assessing the observed UHI humidity amplification effect for making comparisons to the standard atmosphere. Furthermore, the temperature difference WVF model [17] was verified by reproducing the global results of Dessler et al. [29], Gordon et al. [30], and Liu et al. [23]. Liu et al.'s [23] value of 1.6 $\text{Wm}^{-2}\,\text{K}^{-1}$ reproduced the most accurate results according to the temperature difference WVF model (and is also the most recent estimate). Then relative to the background climate average standard atmosphere (see Appendix C.2), the increased microclimate amplification estimate in humid UHIs is about

$$A_{U-WVF} = 3.4\,\text{Wm}^{-2}\text{K}^{-1}/1.6\,\text{Wm}^{-2}\text{K}^{-1} = 2.125 \qquad (9)$$

This model, which is part of Model 4, is supported with alternative data in Section 3.1.10.1. Here $A_{U-WVF}$ is then a ratio of WVF estimates between the local and standard atmosphere observed in UHI humid climates. In humid (denoted as Wet) UHI climates, Equation (8) also applies so that the total amplification is

$$A_{U-Wet} = A_{U-WVF} \times A_{U-Dry} = 2.55 \qquad \textbf{Model 2} \qquad (10)$$

Note that this is below the UHI footprint estimate of 3.2 discussed in Section 2.1. The UHI area footprint amplification estimates incorporate many mechanisms such as loss of evapotranspiration cooling, temperature inversions, aerosols, and so forth. Therefore,

Model 2 is likely conservative and helps to demonstrate the feasibility of Zhang et al.'s [1] observations. We also provide support for this estimate in Section 3.1.10.1.

### 3.1.5. Dry and Humid UHI Mixed Climate Amplification Estimate: Model 3

To be able to apply microclimate UHI amplification estimates, a weighted mixed wet-dry climate assessment is helpful. This will be exemplified in the next section for AHR GW estimates. Cherlet et al. [26] break down the percentages of cities in humid versus dry climates. In terms of climate zones, the population distribution closely follows big cities that are estimated with 33% of the population living in drylands and 67% in non-dryland or humid zones. Interestingly, this closely follows the fact that 71% of the earth's composition consists of oceans.

Given the Cherlet et al. [26] breakdown for dry and humid climates, a mixed dry-humid amplification value due to Models 1 and 2, indicates

$$\overline{A}_{UHI\_mixed} = \left(0.33 A_{U-Dry} + 0.67 A_{U-Wet}\right)1.05 = (0.32 \times 1.2 + 0.68 \times 2.55)1.05 = 2.2 \ \textbf{Model 3} \qquad (11)$$

According to Cherlet et al. [26], the dry region estimate of 33% consists of 11% sub-humid and 15% semiarid, conditions. This amounts to about a 5% increase correction in Equation (11) which is represented by the 1.05 factor. The details for this small 5% correction are provided in Appendix E.

### 3.1.6. UHI Footprint Amplification Estimate: Model 4

Model 4 is based on estimates for the heat flux increase requirements in Section 3.1.10.1 to support the data observations by Zhao et al. [16]. In that section, it is shown that to verify Zhao et al. [16] data observations that UHIs in wet climates are 3.3 K higher than in dry areas, an increase in amplification is required larger than the results of Model 3. This value also is found to help confirm Zhang et al.'s [1] findings adding to modeling self-consistency. This increase is also supported by prior studies [25] that take into account estimates for UHI footprints. The UHI footprint is an indication of amplified heat spreading beyond the city boundaries. A footprint type of amplification estimate, shown in Figure 1, may be conservative. As discussed in Section 2.2 and Figure 1, footprint estimates go as the city areas [33], and assessments show that area growth from 1950–2019, the period of interest, is about 3.2 times, and dome estimates are about a factor of 8.4 [25]. This 3.2 value also matched the footprint assessment by Zhou et al. [33]. Using the lower footprint estimate (Section 2.1) and the results in Section 3.1.10.1 to justify this increase, a footprint (FI) factor is applied to Model 3 of $\overline{A}_{FI} = 1.45$ in Equation (11) yielding the overall UHI footprint amplification ($A_{Footprint}$) estimate

$$A_{Footpring} = \overline{A}_{UHI\_mixed} \times \overline{A}_{FI} = 2.2 \times 1.45 = 3.2 \quad \textbf{UHI Footprint Model 4} \qquad (12)$$

The increase due to $A_{FI} = 1.45$ essentially takes into account Figure 1 additional non-GHG effects such as temperature inversion, aerosol/haze pollution effects, solar canyon issues, and loss of cooling and is shown to be in line with observed data in Section 3.1.10.1.

### 3.1.7. Dry and Humid Rural Mixed Climate Amplification Estimate: Model 5

Geometry strongly differs in rural compared to urban areas which are dominated by cities. In rural settlement areas, microclimate amplification is relatively lighter in contribution from $CO_2$ and water vapor feedback/GHGs for humid, sub-humid, and semiarid areas. However, there are still substantial ISAs in rural regions. Huang et al. [28] estimated that about 67% of ISAs are located in rural settlements. There are still similar amplification contributions even in light settlements besides GHGs such as loss of evapotranspiration cooling and wind reductions. We can anticipate some average amplification (i.e., >1). Therefore, we can provide a very small amplification estimate increase for rural areas, yielding.

$$\overline{A}_{Rural} = 1.1 = \overline{A}_{UHI\_mixed}/2 \qquad \textbf{Rural Model 5} \qquad (13)$$

### 3.1.8. Estimating AHR Influence on Global Warming Using Microclimate Amplification Models

While it is helpful to separate climates in terms of humid versus dry areas, now that we have a mixed Model 3 local amplification estimate value of 2.2 and footprint estimate $\overline{A}_{Footprint} = 3.2$, we can use these to assess the urbanization AHR global warming influence. Per Equations (4), (6), (11), and (12), we can now combine the local microclimate and background global amplification estimate with the AHR assessment in Equation (4). Then the global warming impact due to the AHR with background global and microclimate secondary amplification effects yields

$$\%P_{\text{AHF-GS\&LS}}\,(2021) = \frac{P_{\text{AHF-GS\&LS}}}{5.1\ \text{Wm}^{-2}} = \frac{\left\{0.0356\ \text{Wm}^{-2}(A_{UHI-Mixed} \times A_{FI} \times 0.75 + 0.25 \times A_{Rural})\,2.15 \times 1.62\right\}}{5.1\ \text{Wm}^{-2}} = \begin{cases} 4.7\%\ for\ A_{FI} = 1 \\ 6.5\%\ for\ A_{FI} = 1.45 \end{cases} \quad (14)$$

Here we insert $A_{UHI-Mixed} = 2.2$ from Equation (11), $A_{Rural} = 1.1$ from Equation (13), $A_{FI} = 1$ and 1.45 from Equation (12). The microclimate amplification factors apply mainly to UHI areas where according to the IEA [5] "with 4.2 billion people living in cities today, urban areas account for 75% of global energy consumption" (see also Yang et al. [14]) yielding the 0.75 weighting factor in Equation (14). This result finds estimates of 4.7% to 6.6% GW influence (0.24 Wm$^{-2}$–0.34 Wm$^{-2}$) due to AHR, when both global and microclimate UHI amplifications are considered. Results are highly dependent on the amplification estimates and are independent of the urbanization area.

The results found as much as 6.5% GW influence, yielding a 0.92% decade$^{-1}$ increase; this energy consumption estimated GW effect is from 1950 to 2022 and as expected correlates strongly to the population growth rate (Feinberg, [3]). This is evident from the energy percent increase in 2021 of 1.2% (=2133 TWh/176,431 TWh) which is close to the population growth rate that has decreased in the last 70 years from 2% to 0.89% (https://www.worldometers.info/world-population/, last accessed on 11 January 2023). Since the population growth rate is slowing down, likely, the energy consumption rate may eventually decrease somewhat. However, it is clear from the energy consumption GW effect, that the population growth rate is too high and is still the key issue [3].

### 3.1.9. Area and AHR Local Baseline Heat Flux Estimates

Results were obtained by averaging the anthropogenic warming energy over the entire earth. However, it is likely most of this AHR occurs in cities where 55% of the world's population lives [4]. IEA [5] and Yang et al. [14] estimated that 75% of energy is consumed within cities. In this and the next section, a possible scenario illustrates how modeling estimates can be shown to be self-consistent with the final results in Section 3.2.2 and agree with the main data observation by Zhao et al. [16] and other authors supporting the models.

To estimate heat flux and likely amplification effects it is helpful to overview some of the literature observations. Pigeonet et al. [31] investigated the AHR in the European agglomeration of Toulouse in France. They found that vehicle traffic was the major source of the AHR during summer. During winter, the AHR reached 100 Wm$^{-2}$ in the densest areas, whereas it ranged between 5 and 25 Wm$^{-2}$ in suburban areas. Chen et al. [32] determined, "The annual mean AHR flux in concentrated regions reached 20 W m$^{-2}$ in 2009". Bohnenstengel et al. [8] in London found values in the city center exceed 400 Wm$^{-2}$ and values at the fringes of London are of the order of 60 Wm$^{-2}$ in December. Narumi et al., 2009 studied heat release quantities in a Japanese megacity up to 48 Wm$^{-2}$ and showed that calculated AHR results were similar to observed atmospheric $\Delta T$ values.

To obtain an average AHR baseline urban heat flux estimate, it is helpful to take a lower value. Here we will use a possible scenario for a low AHR baseline heat flux of 6.5 Wm$^{-2}$. The reason to select this value will be clear in Section 3.1.10.1. Its absolute value is not the main goal as estimates focus on difference data in Section 3.1.10.1. However, the value simplifies difference data estimates and shows self-consistency in models and results. It is also close to the Pigeonet et al. [31] 5 Wm$^{-2}$ lower measurement. It is reasonable to

use a low and likely unamplified heat flux baseline estimate. This is then the target goal in Equation (15).

To obtain an idea of how 6.5 Wm$^{-2}$ could occur we can consider Equation (4) results for an estimate of the increase in anthropogenic waste heat flux in UHIs. To do this, we need an estimate of the UHI area. Feinberg [25,53] using the definition of what is urban by Liu et al. [54], showed the portion of the earth that has been urbanized in this time period is 0.28%. Other authors Zhao et al. [55], for example, reported about 0.2% in 2020. Huang et al. [28] reported 0.994 million km$^2$ (0.2% of the earth) using 2016 mappings.

Sun et al. [27] estimated impervious surface areas where "we created spatially explicit and temporally consistent global maps of impervious surface area at 10-m spatial resolution for 2015 and 2018, namely the high-resolution global impervious surface area (Hi-GISA) maps . . . . The total area of global impervious surfaces grew from 1.27 million km$^2$ in 2015 to 1.29 million km$^2$ in 2018" (about 0.255% of the earth).

Theobald et al. ([56] questioned the amount of earth that has been transformed by human modification from 1990 to 2017. Theobald et al. [56] made two key estimates: "between 1990 and 2015 resulted in 1.6 million km$^2$ of natural land lost, and a contemporary ($\sim$2017) estimate of human modification that included additional stressors and found that globally 18.5 million km$^2$ of lands have been modified. However, a breakdown for impermeable surfaces was not easily estimated from their data but shows concern for higher estimates.

We see that there is a wide range of areas of the earth that is modified and subject to interpretation regarding urbanization. It is of interest to average the value of Equation (4) over the urbanization areas. Therefore, in this assessment, we can look at several urbanization area values (see Tables 4 and 5).

As an initial calculation, first, consider the urbanization estimate of 0.255% due to Sun et al.'s [27] for ISAs. We use an adjusted estimate from Huang et al. [28] of about 40% for ISAs in urban areas that are likely dominated by UHI building sides (see Section 3.2.1.1 discussion). This is a 0.4-factor multiplier in this area. The actual UHI built-up areas for AHR are larger when one takes into account tall buildings with small multiple rooms of apartments and office spaces each dissipating heat fluxes due to heating and cooling energy use. We can account for this in Equation (15) with an f factor. Here an adjustment factor of $f = 4.0$ is used as the area increase which provides the desired scenario for the baseline values P$_{\text{AHR-UHI}}$ = 6.5 Wm$^{-2}$. In general, exact estimates are not a major concern. This rough approximation will provide temperature difference estimates in the next section which are not very affected by the absolute value. As we referenced earlier, 75% of AHR energy is consumed in cities (IEA [5] and Yang et al. [14]). We can then incorporate this energy factor in the numerator and area factors in the denominator to obtain a possible scenario for the baseline heat flux

$$P_{\text{AHF}-\text{UHI}}(2021) = 0.0356\,\text{Wm}^{-2} \times 0.75/(0.00255 \times f \times 0.4) = 6.5\,\text{Wm}^{-2} \quad (15)$$

Note that averaging over smaller areas would cause a substantial increase in anthropogenic heat flux estimates in Equation (15). Other areas are given in Tables 4 and 5. The goal here is not to obtain exact values but to provide an idea of a possible scenario illustrating how this heat flux could occur and obtain AHR $\Delta T$ difference data feasible attribution estimates in the next section which are not very effected by the accuracy in Equation (15) as will be shown. This will help support amplification modeling and their estimates as we will see in Section 3.1.10 (which will be further supported by humidity data from Zhao et al. [16] and other authors as detailed in Sections 2.2 and 3.1.4 and Appendix D).

### 3.1.10. Supporting Amplification Models 1–4 in Dry and Wet Environments with UHI $\Delta T$s Estimates

We can estimate $\Delta T_{\text{UHI}}$ attribution values due to AHR in dry and wet environments using amplified baseline values and help support modeling and their values as well. Consider the possible baseline heat flux scenario in Equation (15) estimate of 6.5 Wm$^{-2}$. Although

this is not an exact estimate, $\Delta T$ attribution is based mainly on amplification estimated according to Equation (A1) and differences which are not very affected by Equation (15) absolute value. Using the resulting value, the estimated dry Model 1 amplification 1.2-factor (Equation (8)) provides an assessment of the increase in the baseline heat flux for Equation (15) due to local GHGs in UHI areas as

$$1.2 \times 6.5 \, \text{Wm}^{-2} = 7.8 \, \text{Wm}^{-2} \tag{16}$$

Such values, for example, $\Delta \overline{P}_U = 7.8 \, \text{Wm}^{-2}$ are not unreasonable as discussed in the prior section.

From Appendix A, Equation (A2), we find that $\Delta T_{AHR-UHI\_Dry} = 1.3 \, °C$ where

$$\Delta T_U = \left[ \frac{\left( \Delta \overline{P}_{AHF\_UHI} + \sigma(\overline{T})^4_{Ambient} \right)}{\varepsilon_{eff} \sigma} \right]^{1/4} - \overline{T}_{Ambient} = 1.3 \, °C \tag{17}$$
$$\Delta \overline{P}_{AHF\_UHI} = 7.79 \, \text{Wm}^{-2}, \overline{T}_{Ambient} = 24 \, °C$$

This result is shown in the next to last column in Table 4. For the baseline estimate (Equation (15)) we find that $\Delta T_U = 1.09 \, °C$ using Equation (A2) where

$$\Delta T_U = \left[ \frac{\left( \Delta \overline{P}_{AHF\_UHI} + \sigma(\overline{T})^4_{Ambient} \right)}{\varepsilon_{eff} \sigma} \right]^{1/4} - \overline{T}_{Ambient} = 1.085 \, °C \tag{18}$$
$$\Delta \overline{P}_{AHF\_UHI} = 6.5 \, \text{Wm}^{-2}, \overline{T}_{Ambient} = 24 \, °C$$

Comparing Equations (17) and (18), results in this assessment indicate that AHR creates a temperature increase in UHIs, this is the baseline result yielding a 1.09 °C rise according to Equation (18). Then GHGs raise this to 1.3 °C as found in Equation (17) yielding a difference of 0.215 °C increase. Therefore, GHGs are responsible for 17% (=0.215/1.3) of the temperature increase in Equation (17) (shown in Col. K, Table 4). Note the stability of the last column percentages even though the baseline data is varying in Col. 3 illustrating the insensitivity of difference data to the absolute values in Col. 3 (i.e., Equation (15)).

These $\Delta T$ temperature increases due to GHG and WVF attribution effects in dry and wet climates are shown in Col. J in Table 4 with different areas.

**Table 4.** Physics-based model estimates for dry (Model 1) and humid (Model 2) climates.

| ISA %Areas * | Climate | Baseline $P_{AHR-UHI}$ Wm$^{-2}$ Equation (15) | Baseline $\Delta \overline{T}_{UHI}$ °C Equation (17) | Dry Amp. Equation (8) | Wet Amp. Equation (9) | $A_U$ Wet & Dry Combined E × F | AHR $P_{AHR-LS}$ Wm$^{-2}$ D × G | Estimate AHR $\Delta \overline{T}_{UHI}$ °C Equation (16) | GHG-WVF Amp. Au $\Delta T$ Percent Effect |
|---|---|---|---|---|---|---|---|---|---|
| **A** | **B** | **D** | | **E** | **F** | **G** | **H** | **J** | **K** |
| | | | | Model 1 (Dry) | | | | | |
| 0.22 | **Dry** | 7.53 | 1.26 | 1.2 | 1 | 1.2 | 9.03 | <u>1.5</u> | <u>17%</u> |
| 0.255 | **Dry** | 6.5 | 1.085 | 1.2 | 1 | 1.2 | 7.79 | <u>1.3</u> | <u>17%</u> |
| 0.33 | **Dry** | 5.02 | 0.84 | 1.2 | 1 | 1.2 | 6.02 | <u>1.01</u> | <u>17%</u> |
| | | | | Model 2 (Wet) | | | | | |
| 0.22 | Humid | 7.53 | 1.26 | 1.2 | 2.125 | 2.55 | 19.2 | <u>3.17</u> | <u>60%</u> |
| 0.255 | Humid | 6.5 | 1.09 | 1.2 | 2.125 | 2.55 | 16.6 | <u>2.74</u> | <u>60%</u> |
| 0.33 | Humid | 5.02 | 0.84 | 1.2 | 2.125 | 2.55 | 12.8 | <u>2.13</u> | <u>61%</u> |

* Estimate UHI area with building area increases (see Equation (15)).

Using the method of Equation (11), we can estimate the mixed climate estimate for the last column for *GHG-WVF* attribution finding

$$A_{GHG-WVF}(\Delta T) = (0.33 \times 17\% + 0.67 \times 60\%)1.05 = 48\% \tag{19}$$

This result suggests the weighted average temperature increase attributed to *GHG* or *WVF* is about 48%.

First notice the difference in dry climates between the estimated AHR $\Delta T_{\text{UHI}}$ and the baseline $\Delta T_{\text{UHI}}$: $1.5 - 1.26\,°C = 0.25\,°C$, $1.3 - 1.09\,°C = 0.216\,°C$, and $1.01 - 0.85\,°C = 0.167\,°C$. This averages 0.21 °C due to Model 1 GHG (mainly $CO_2$ increase). In Appendix D, a study by Liu et al. [57] is discussed related to the COVID lockdown period that studied over 300 megacities in China where traffic was minimal and showed an average day-night reduction in surface temperature of $0.24 \pm 0.22$ K and the canopy of $0.41 \pm 0.27$ K representing about a 25% and 40% reduction, respectively. We see that surface temperature is likely more related to $CO_2$ GHGs while the canopy is likely more related to AHR and the GHG effect. The surface temperature difference averaging 0.21 °C provides a nice level of confidence in the Model 1 estimate which is used in Table 2 for the dry climate with the 1.2 factor related to $CO_2$ and yielding a somewhat lower estimate compared to Liu et al. [57] surface temperature change observation of 0.24 °C, due to a likely reduction in $CO_2$ emission because of lockdown.

Next note the differences in $\Delta T$s between dry and wet climates: $3.17 - 1.5\,°C = 1.67\,°C$, $2.74 - 1.3\,°C = 1.44\,°C$, and $2.13 - 1.01\,°C = 1.12\,°C$. This averages 1.41 °C. Zhao et al. [16] observed on average that wet climates were 3.3 K higher than dry climates. This indicates modeling is not completed yet and we can improve estimates as detailed in the next section.

3.1.10.1. Further Supporting Models 1–4 in Dry and Wet Environments

There are two main reasons that the dry-wet climate difference was only 1.2 °C in Table 4, yielding 2.1 °C lower than what Zhao et al. [16] observed. In Column G, the amplification values $A_u$ could be higher. Another key reason is that $\Delta T$ in dry and wet climates also depends on solar heating of ISAs in UHIs. We can provide a likely scenario to make up for the missing heat fluxes from solar heating of ISAs in UHIs in Tables 5 and 6 by increasing the baseline estimate proportionately to the anticipated increase due to global warming percentages found in Table 7 bottom row. This will aid in demonstrating self-consistency in modeling alignment with data. The additional GW% from urban ISAs is 4% (bottom row, Col. 10, Table 7). Note also in the bottom row the 6.5% value (Col. 8, Table 7 where 6.5% goes as 6.5 $\text{Wm}^{-2}$). This prompted the simple proportionate value of 6.5 $\text{Wm}^{-2}$ used in Equation (15). Now we are in a position to add another proportionate baseline value in Col. D in Table 4 of 4 $\text{Wm}^{-2}$ (Col. 10, Table 7 where 4% goes as 4 $\text{Wm}^{-2}$) to each value in Col. D Table 4, yielding the Table 5 Col. D values. That is for self-consistency with ISA and AHR global warming data, we can provide proportionate percentages to heat fluxes values. For example, the total baseline heat flux Equation (15) would increase to

$$6.5\,\text{Wm}^{-2} + 4\,\text{Wm}^{-2} = 10.5\,\text{Wm}^{-2} \tag{20}$$

shown in data Row 5, Table 5. We next increase the amplification factors by $A_{FI} = 1.45$ factor in Equation (12), Model 4 to illustrate its effect and how it is justified in this scenario. We can see that it will help to simulate the Zhao et al. [16] observations using the $A_{FI}$ factor. This is carried out in Table 5.

**Table 5.** Physics-based model estimates for dry (Model 1) and humid (Model 2) climates.

| ISA% Areas | Climate | Baseline $P_{\text{AHR-UHI}}$ $\text{Wm}^{-2}$ Equation (19) | Baseline $\Delta\overline{T}_{UHI}$ °C Equation (17) | Dry Amp. Equation (8) | Wet Amp. Equations (9) and (12) x$A_{FI}$ = 1.45 | $A_U$ Wet & Dry Combined E × F | $P_{\text{AHR-LS}}$ $\text{Wm}^{-2}$ D × G | $\Delta\overline{T}_{UHI}$ °C Equation (16) | GHG-WVF Amp. $A_u \Delta T$ Percent Effect |
|---|---|---|---|---|---|---|---|---|---|
| **A** | **B** | **D** | | **E** | **F** | **G** | **H** | **J** | **K** |
| | | | | Model 1 (Dry) | | | | | |
| 0.22 | Dry | 11.53 | 1.92 | 1.2 | 1 × 1.45 | 1.74 | 20.1 | 3.31 | 42% |
| 0.255 | Dry | 10.5 | 1.75 | 1.2 | 1 × 1.45 | 1.74 | 18.3 | 3.02 | 42% |
| 0.33 | Dry | 9.02 | 1.50 | 1.2 | 1 × 1.45 | 1.74 | 15.7 | 2.61 | 42% |
| | | | | Model 2 (Wet) | | | | | |
| 0.22 | Humid | 11.53 | 1.92 | 1.2 | 2.125 × 1.45 | 3.7 | 42.6 | 6.92 | 72% |
| 0.255 | Humid | 10.5 | 1.75 | 1.2 | 2.125 × 1.45 | 3.7 | 38.8 | 6.32 | 72% |
| 0.33 | Humid | 9.02 | 1.50 | 1.2 | 2.125 × 1.45 | 5.7 | 33.4 | 5.45 | 72% |

Note now the differences in $\Delta T$s between dry and wet climates are $6.92 - 3.31$ °C = 3.6 °C, $6.32 - 3.02$ °C = 3.29 °C, and $5.45 - 2.61$ °C = 2.85 °C. The average of the differences is 3.25 °C which is very close to what Zhao et al. [16] observed that UHIs in wet climates are 3.3 K higher than UHIs in dry climates. Furthermore, note the 0.255% area value provides the best results, this somewhat supports Sun et al.'s [27] area estimate. Table 5 results are due to the added likely scenarios in heat flux from unshaded ISAs and increasing the amplification by $A_{FI}$ = 1.45 the Model 4 factor in Equation (12). Note also that the likely scenarios of $\Delta T$ values are not uncommon for UHI observations. Indeed, a study by climate central [58] found in 159 cities in the US, that complex UHI could be much higher with 8–11 °C $\Delta T$ values. The values in Table 5 then illustrate likely possible data scenarios.

This helps justify Equation (12), Model 4 footprint assessment as well it supports Model 1 and 3 amplification values by finding reasonableness with data observation including Zhao et al. [16] and self-consistency in results in Table 7 since these models are also used in Equation (14) and Tables 6 and 7. Given the difficulty of providing physics-based amplification estimates, such data-aided verification is extremely helpful in proving modeling confidence.

*3.2. Part II: Impermeable Surfaces Albedo Land Cover Change Effects*

Considering the AHR GW% estimate ranging from 4.7% to 6.5% averaging 5.6% in Equation (14), the discrepancy with Zhang et al. [16] assessment is

$$12.7\% - 5.6\% = 6.9\% \tag{21}$$

Therefore, about 6.9% yields a leftover urbanization global warming likely uncertainties due to other issues, primarily, the urbanization land cover albedo effect (Feinberg, 2020), with various contributions differing from city to city. This is the topic of Part II.

3.2.1. GW Effect due to Solar Heating of ISAs with Secondary Effects

It is common knowledge and many papers have reported high land surface contact temperatures (LSCT) due to solar heating of roads and roofs. Rasmussen et al. [59] reported results of NASA ECOSTRESS found in Las Vegas pavement areas where temperatures were up to 27 °C hotter than non-ISA. Pavements are often so hot they cause health risks in contact burns (Kowal-Vern et al. [60]) as well hot pavements cause problems for dogs [61]. Interestingly, air temperatures quickly become uncorrelated with surface temperatures at an altitude of 5 feet. Morabito et al. [62] estimated about 0.4 °C to 1 °C air temperature rise for every 20% increase in impermeable surfaces near building areas. Many authors [1,63] measure the surface UHI $\Delta T$ as $\Delta T_{\text{Surface UHI}} = \text{LSCT}_{\text{built-up}}\text{-LSCT}_{\text{rural}}$. However, contact measurements on high-temperature pavements (>5 °C) between rural and built-up areas can be similar to the ambient differences.

Estimates are provided for albedo and LSCT of impermeable surface areas due to solar heating and the solar 'energy in' absorbed. Using the LSCT method is helpful as UHIs in megacities have large solar ISAs on building sides with windows and solar canyon effects that satellite albedo measurements are unable to assess and are not considered in albedo mappings (see Section 4.2). 'Energy in' is a key factor in assessing the ISA for urbanization energy flux. In this approach, we can study solar heating of unshaded ISA.

In this section, impermeable surface forcing is assessed using contact temperature difference ($C\Delta T$) with albedo estimates (see Table 6). LSCT assessments above ambient, are a helpful practical additional ISA metric that we are familiar with as pavement temperatures can become very high.

Mentaschi et al. [62] studied surface urban heat island temperatures and found, "Within many cities, there are hotspots where extreme surface UHI intensity is 10–15 °C higher compared to relatively cooler city parts." This is mild compared to the 28 °C difference estimated in the NASA observation [59] and 32 °C by Knox [64] It is also easy to measure, the author [65] using a heat gun observed older asphalt road temperatures in suburban areas were about 51 °C and new roads at about 53 °C taken at an ambient

temperature of 33 °C and concrete about 44 °C. This is 20 °C higher than the ambient for asphalt and 11 °C higher for ]. One may observe that new road albedos may not age as much as old roads since surfaces appear smoother so they likely clean better in the rain. This is unfortunate as the surfaces are likely to maintain higher temperatures compared to older roads. Note the difference that concrete is 13 °C cooler than asphalt. At cooler ambient of 12 °C, results found that pavements were 7 °C to 10 °C higher than ambient depending on the time of day and if the streets were surrounded by buildings.

To help determine the impermeable surface energy dissipated from solar heating, we can use the global ISA estimated by Sun et al. [27] as discussed in Section 3.1.9 found 1.29 million km$^2$ (~0.255%) in 2018.

### 3.2.1.1. Irradiance and ISA Percentages Adjustment Modeling Factors

To adjust for UHI solar irradiance (F-Solar UHI) and urban ISA percentages, several factors apply where

$$\text{F} - \text{Solar UHI} = \text{f1} \times \text{f2} \times \text{f3} = 0.47 \times 0.65 \times \begin{cases} 0.33 \\ 0.41 \\ 0.5 \end{cases} = \begin{cases} 0.1008 \\ 0.1253 \\ 0.1528 \end{cases} \quad \textbf{UHI Factor} \tag{22}$$

The factors are as follows: f1 = 0.47 accounts for the global average irradiance through clouds [24], and f2 = 0.65 accounts for the unshaded UHI areas which are higher than 50% to account for the solar canyon effect. The shade factor is discussed below. Lastly, the value f3 is the fraction of urban ISA likely dominated by city building sides. Huang et al. [28] report ISA-weighted averages of 33% urban and 67% rural. The urban ISA seems underweighted considering that there are 540 million buildings in the world as tagged in the OSM database (https://taginfo.openstreetmap.org/keys, last accessed on 29 December 2022). Huang et al. (2022) use a complex algorithm for building sides and provided assessment errors (also see Section 4.2). We might best use a variety of values to adjust for the possible underweighted error. Additionally, the definition of urban and rural is not easy to define. Therefore, within the Huang et al. [28] margin of error for urban ISAs, we consider values 50%, 41%, and 33% for the urban ISA percent as shown in Equation (22).

Outside the UHI area, an adjustment factor is needed for rural irradiance and ISA percentage estimates, similar to Equation (22) as follows

$$\text{F} - \text{Solar UHI} = \text{f1} \times \text{f2} \times \text{f3} = 0.47 \times 0.75 \times \begin{cases} 0.67 \\ 0.59 \\ 0.5 \end{cases} = \begin{cases} 0.2362 \\ 0.208 \\ 0.1763 \end{cases} \quad \textbf{Rural} \tag{23}$$

The rural factors are as follows: f1 = 0.47 accounts for the global average irradiance through clouds ([24] and f2 = 0.75 accounts for unshaded areas where it is anticipated that less shade occurs compared to UHI tall building effects. Lastly, we use a variety of ISA rural area fractions for f3 in Equation (23) based on Equation (22) values (1-f3$_{\text{urban}}$).

### 3.2.1.2. Global Warming Estimates due to Solar Heating of ISAs

Table 6 provides global warming (GW) influences at various LSCT changes denoted as C$\Delta T$, above ambient (of 14.5 °C) increase in Col. A. The actual LSCT is then 14.5 °C + C$\Delta T$. Initial values in this table use the average ISA estimates of 41% urban and 59% rural (Equations (22) and (23)). Table 7 provides the alternate values in Equations (22) and (24). An example calculation for Table 6 is provided in the next section.

Note the albedo estimate (Col. B) from the LSCT in Col. A. At 0 °C above ambient, the albedo is 0.3 which is taken as the average reflectivity of the earth's system and this requires we use the average surface temperature of the earth of 14.5 °C for the albedo reference point (see Equation (25)). It is important to note several key albedo estimates of 0.15, 0.133, and 0.115 are highlighted. These are close to values reported by Sugawara and Takamura [66] of 0.12 for the cities in Japan they investigated, Tricia et al. [67] also found a

value of 0.12 for UHI urbanization Boston areas. As well Guo et al. [19] reported average albedo values in the range of 0.15 in China.

Consider the LSCT above ambient value CΔT of 9 °C (albedo value of 0.15). The upper area uses a local amplification factor from Model 3 of 2.2. This yields a 2.4% urban global warming influence dominated by cities. We would anticipate that rural ISAs have higher average LSCT as they have about 20% more asphalt roads [28]. Therefore, we selected an increased CΔT of 10 °C (albedo of 0.133) yielding a rural contribution of 2.3%. Then the total ISA contribution is 4.7% (see data row 4 in Table 7) over the approximate time period of 1950 to 2019. Table 7 provides an estimate for different ISA areas, urban-to-rural area ratios, and local amplification estimates.

**Table 6.** Effect of solar heating with estimates of ISTs and its likely global warming impact.

| LSCT CΔT Average above Ambient 14.5 °C | Albedo Average Estimate At LSCT * Equation (A7) | Solar Heating ISA Forcing Wm⁻² Equation (A1) | Total Solar Heating ISD Watts (Using ISA $1.3 \times 10^6$ km²) | Col. D Div by Area of Earth Wm⁻² | Col. E Div by 5.1 Wm⁻² %GW As in Equation (5) | Global Amp Factor $1.62 \times 2.15$ Table 3 | F-Solar UHI | F-Solar Rural | Local Amp. Factor Equations (11) and (12) | Local Rural Amp $A_{Rural}$ Equation (13) | UHI Global % F × G × H × J | Rural %GW F × G × I × K |
|---|---|---|---|---|---|---|---|---|---|---|---|---|
| **A** | **B** | **C** | **D** | **E** | **F** | **G** | **H** | **I** | **J** | **K** | **L** | **M** |
| | | | | GHG-WVF Amplification (Col. J) | | | | | | | | |
| 0 | 0.300 | 0.000 | 0.000 | 0.000 | 0.00% | 3.48 | 0.1253 | 0.208 | 2.2 | 1.1 | 0.0% | 0.0% |
| 1 | 0.284 | 5.43 | $7.05 \times 10^{12}$ | 0.014 | 0.27% | 3.48 | 0.1253 | 0.208 | 2.2 | 1.1 | 0.3% | 0.2% |
| 5 | 0.219 | 27.70 | $3.60 \times 10^{13}$ | 0.071 | 1.38% | 3.48 | 0.1253 | 0.208 | 2.2 | 1.1 | 2.3% | 1.1% |
| 8 | 0.168 | 45.02 | $5.85 \times 10^{13}$ | 0.115 | 2.25% | 3.48 | 0.1253 | 0.208 | 2.2 | 1.1 | 2.2% | 1.8% |
| **9** | **0.150** | **50.91** | $\mathbf{6.62 \times 10^{13}}$ | **0.130** | **2.54%** | **3.48** | **0.1253** | **0.208** | **2.2** | **1.1** | **2.4%** | **2.0%** |
| **10** | **0.133** | **56.87** | $\mathbf{7.39 \times 10^{13}}$ | **0.145** | **2.84%** | **3.48** | **0.1253** | **0.208** | **2.2** | **1.1** | **2.7%** | **2.3%** |
| 11 | 0.115 | 62.88 | $8.17 \times 10^{13}$ | 0.160 | 3.14% | 3.48 | 0.1253 | 0.208 | 2.2 | 1.1 | 3.0% | 2.50% |
| 12 | 0.097 | 68.95 | $8.96 \times 10^{13}$ | 0.176 | 3.45% | 3.48 | 0.1253 | 0.208 | 2.2 | 1.1 | 3.3% | 2.74% |
| 13 | 0.079 | 75.08 | $9.76 \times 10^{13}$ | 0.191 | 3.75% | 3.48 | 0.1253 | 0.208 | 2.2 | 1.1 | 3.5% | 2.90% |
| 14 | 0.061 | 81.28 | $1.06 \times 10^{14}$ | 0.207 | 4.06% | 3.48 | 0.1253 | 0.208 | 2.2 | 1.1 | 3.9% | 3.2% |
| | | | | Footprint Amplification (Col. J) | | | | | | | | |
| 0 | 0.300 | 0.000 | 0.000 | 0.000 | 0.00% | 3.48 | 0.1253 | 0.208 | 3.2 | 1.1 | 0.0% | 0.0% |
| 1 | 0.284 | 5.43 | $7.05 \times 10^{12}$ | 0.014 | 0.27% | 3.48 | 0.1253 | 0.208 | 3.2 | 1.1 | 0.4% | 0.2% |
| 5 | 0.219 | 27.70 | $3.60 \times 10^{13}$ | 0.071 | 1.38% | 3.48 | 0.1253 | 0.208 | 3.2 | 1.1 | 1.9% | 1.1% |
| 8 | 0.168 | 45.02 | $5.85 \times 10^{13}$ | 0.115 | 2.25% | 3.48 | 0.1253 | 0.208 | 3.2 | 1.1 | 3.1% | 1.8% |
| **9** | **0.150** | **50.91** | $\mathbf{6.62 \times 10^{13}}$ | **0.130** | **2.54%** | **3.48** | **0.1253** | **0.208** | **3.2** | **1.1** | **3.5%** | **2.0%** |
| **10** | **0.133** | **56.87** | $\mathbf{7.39 \times 10^{13}}$ | **0.145** | **2.84%** | **3.48** | **0.1253** | **0.208** | **3.2** | **1.1** | **4.0%** | **2.3%** |
| **11** | **0.115** | **62.88** | $\mathbf{8.17 \times 10^{13}}$ | **0.160** | **3.14%** | **3.48** | **0.1253** | **0.208** | **3.2** | **1.1** | **4.4%** | **2.50%** |
| **12** | **0.097** | **68.95** | $\mathbf{8.96 \times 10^{13}}$ | **0.176** | **3.45%** | **3.48** | **0.1253** | **0.208** | **3.2** | **1.1** | **4.8%** | **2.74%** |
| 13 | 0.079 | 75.08 | $9.76 \times 10^{13}$ | 0.191 | 3.75% | 3.48 | 0.1253 | 0.208 | 3.2 | 1.1 | 5.1% | 2.90% |
| 14 | 0.061 | 81.28 | $1.06 \times 10^{14}$ | 0.207 | 4.06% | 3.48 | 0.1253 | 0.208 | 3.2 | 1.1 | 5.7% | 3.2% |

* LSCT = 14.5 °C + CΔT.

### 3.2.1.3. Example Calculation

As an example of the calculations in Table 6, Consider the highlighted footprint data row in Col. A where CΔT = 10 °C (data row 16), yielding an LSCT temperature of 14.5 °C + 10 °C = 24.5 °C. From Appendix A, Equation (A1), we find the results in Col. C of

$$\Delta \overline{P}_{Forcing} = \varepsilon_{eff}\sigma\left(\left(\overline{T}_{Ambient} + \Delta T_U\right)^4 - \left(\overline{T}\right)^4_{Ambient}\right) =$$
$$\sigma\left(\left(14.5 + 10 + 273.15\right)^4 - \left(14.5 + 273.15\right)^4\right) = 56.86 \, \text{Wm}^{-2} \tag{24}$$

In Column A, a CΔT of 10 °C yields the albedo value using Appendix B, Equation (A7) estimated as

$$(\Delta\alpha) = \varepsilon_{eff}\sigma\left(\overline{T}_S^4 - \left(\overline{T}_S + \Delta T\right)^4\right)/\frac{S_o}{4} = 56.9/340.25 = 0.167 \tag{25}$$

where $\alpha_S = 0.3 - \Delta\alpha = 0.3 - 0.167 = 0.133$. Note Equation (25) suggests a 10 °C variation for an albedo change of 0.167 or about 0.0167/°C. Heat gun observations from the author (Section 3.2.1) found a 13 °C change between the old concrete and old asphalt. The differences in albedo vary depending on aging and materials, roughly between old asphalt

and concrete is $\Delta\alpha = 0.18$–$0.21$ . This would yield about $0.014/°C$ to $0.0162/°C$. This is fairly close to theory. Note that $\varepsilon_{eff} = 1$ was used and a better value is closer to 0.95 which would yield a small correction in the tables.

We can use the urbanization (urban and rural) area due to Sun et al. [27] in Col. D, $56.9~Wm^{-2} \times 1.3 \times 10^{12} m^{-2} = 7.39 \times 10^{13}$ Watts, dividing this by the area of the earth yields a Col. E value of $0.145~Wm^{-2}$; then dividing this by $5.1~Wm^{-2}$ (see Equation (7)) yields 2.84% in Col F. We then multiply this by the background global amplification factor of $2.15 \times 1.62$, the microclimate amplification factor of 3.2 (Equation (12)), the solar UHI adjustments irradiance factor of 0.1253 (Equation (22)) in Col. H yielding Col. L results of (data row 20)

$$\text{Urban estimate} = 2.84\% \times 2.15 \times 1.62 \times 3.2 \times 0.1253 = 4.0\% \tag{26}$$

However, the rural average unshaded LSCT is estimated to be higher with more open asphalt roads of 11 °C, this row yields 3.14% before applying factors which then finds (data row 21, Table 6)

$$\text{Rural Estimate} = 3.14\% \times 2.15 \times 1.62 \times 1.1 \times 0.208 = 2.5\% \tag{27}$$

Then the total estimate has an average GW influence of about 6.5%. This value is given in the eighth data row in Table 7. Huang et al. [28] reported that roads consist of about 14% of the ISA. This would indicate that about $6.5\% \times 0.14 = 0.91\%$ of global warming is possibly due to roads (shown in Table 7). However, the likely value of 1.1% (see next section) is discussed in the next section. Other estimates are provided in the next section.

In this calculation, the adjustment factors for shade are not well known but estimates provided help in understanding feasibility. This assessment is less straightforward than the estimate for AHR influence in Equation (14) because of the adjustment factors in Equations (22) and (23). However, these factors are believed to be reasonable. Such modeling is likely more conservative than climate models related to GHGs such as $CO_2$ estimates on GW that do not include UHF contributions as detailed in this study (see Appendix F).

Note the 6.5% GW influence is the best estimate and helps assess the root causes of Zhang et al. [1] result (Table 7).

It is interesting to note that Guo et al. [19] in China found in 10 of the 11 cities they assessed had albedo values that averaged around 0.15–0.16. They concluded the cities had an overall cooling effect. By comparison, the 0.15 albedo value in Table 6 yields on a global scale urban value of about a 3.5% GW influence. Note in this study comparisons are made by assessing ISA temperatures that are above ambient. This allows us to work relative to earth ambient estimates. Results show the need to add city amplification effects when deducing land cover albedo influences.

### 3.2.2. Combined Results

From Table 7, the likely scenario for high-temperature unshaded impermeable surfaces (including building sides) averages around 10 °C for urban and 11 °C for rural, above a 14.5 °C ambient (with albedo ISA estimates of 0.133 and 0.115 respectively). This likely scenario considers some of the reported city albedos in the range of the estimates of 0.115 to 0.16 yielding a total UHF contribution of 13% that matches the Zhang et al. [1] assessment. This yields a measure of modeling self-consistency along with the data-aided modeling results in Section 3.1.10. Using these averages, the ISAs are estimated to contribute 6.5% to global warming when considering the footprint 3.2 and rural 1.1 amplification estimates (see last row Table 7). This is broken down with urban ISA contributing an average of 4.0% and rural areas contributing about 2.5% GW influence.

**Table 7.** Combined results from Table 6 with various values.

| UHI LSCT CΔ*T* (°C) above Ambient (%ISA) | UHI ISA Albedo Value | Rural LSCT CΔ*T* (°C) above Ambient (%ISA) | ISA Albedo Rural Values | Local Amp UHI | Rural Local Amp | GW% from AHR | GW% from All ISA | GW% from ISA of Roads | Urban ISA GW% (Rural ISA GW%) | Total ISA & AHR GW% |
|---|---|---|---|---|---|---|---|---|---|---|
| **GHG-WVF Amplification** | | | | | | | | | | |
| 10 (50%) | 0.133 | 11 (50%) | 0.115 | 2.2 | 1.1 | 4.74 | 5.4 | 0.76 | 3.3 (2.1) | 10.1 |
| 10 (41%) | 0.133 | 11 (59%) | 0.115 | 2.2 | 1.1 | 4.74 | 5.2 | 0.73 | 2.7 (2.5) | 9.95 |
| 10 (33%) | 0.133 | 11 (67%) | 0.115 | 2.2 | 1.1 | 4.74 | 5.0 | 0.7 | 2.2 (2.84) | 9.8 |
| **Footprint Amplification** | | | | | | | | | | |
| 9 (50%) | 0.15 | 10 (50%) | 0.133 | 3.2 | 1.1 | 6.5 | 6.2 | 0.87 | 4.3 (1.9) | 12.7 |
| 9 (41%) | 0.15 | 10 (59%) | 0.133 | 3.2 | 1.1 | 6.5 | 5.8 | 0.81 | 3.5 (2.3) | 12.3 |
| 9 (33%) | 0.15 | 10 (67%) | 0.133 | 3.2 | 1.1 | 6.5 | 5.5 | 0.77 | 2.9 (2.6) | 12.0 |
| 10 (50%) | 0.133 | 11 (50%) | 0.115 | 3.2 | 1.1 | 6.5 | 7.1 | 1.0 | 5.0 (2.1) | 13.6 |
| **10 (41%)** | **0.133** | **11 (59%)** | **0.115** | **3.2** | **1.1** | **6.5** | **6.5** | **0.91** | **4.0 (2.5)** | **13.0** |
| 10 (33%) | 0.133 | 11 (67%) | 0.115 | 3.2 | 1.1 | 6.5 | 6.14 | 0.86 | 3.3 (2.84) | 12.6 |
| 11 (50%) | 0.115 | 12 (50%) | 0.097 | 3.2 | 1.1 | 6.5 | 7.6 | 1.1 | 5.3 (2.3) | 14.1 |
| 11 (41%) | 0.115 | 12 (59%) | 0.097 | 3.2 | 1.1 | 6.5 | 7.1 | 1.0 | 4.4 (2.7) | 13.6 |
| 11 (33%) | 0.115 | 12 (67%) | 0.097 | 3.2 | 1.1 | 6.5 | 6.6 | 0.92 | 3.5 (3.1) | 13.1 |
| **Road Estimate** | | | | | | | | | | |
| 12 (41%) | 0.097 | 12 (59%) | 0.097 | 3.2 | 1.1 | NA | 7.6 | 1.1 * | 4.8 (2.7) | NA |
| **Footprint Average Results** | | | | | | | | | | |
| **10 (41%)** | **0.133** | **11 (59%)** | **0.115** | **3.2** | **1.1** | **6.5** | **6.5** | **1.1 *** | **4.0 (2.5)** | **13.0** |

\* Evaluated at an albedo of 0.1 (see Huang et al., 2022).

Asphalt roads ISAs are estimated to have about a 1.1% global warming influence evaluated at an albedo of about 0.1 (next to the last row, Table 7). Note however, the ISA for roads is based on Huang et al.'s [28] estimate, which they detail is likely biased owing to incomplete data and lack of detection on narrow roads due to limitations of the spatial resolution. Therefore, the value in Table 7 that yields a GW influence of 1.1% may be higher.

As it was pointed out these results are not as simple as the AHR assessment in Equation (14) due to the estimates in Equations (22) and (23). Here we are reminded that climate modeling is not a perfect science and the goal here is to provide likely scenarios to illustrate support for the Zhang et al. [1] findings.

Primarily, these results help explain root causes and provide a level of confirmation for the Zhang et al. [1] results found from ground-based station temperature measurements. That is when the ISA and AHR heat flux contributions are added together the GW contribution averages about 13%. Zhang et al [1] state "the magnitude of the urbanization effect on global land averaged annual mean surface air temperature change over 1951–2018 corresponds to urbanization contribution of 12.7%". The Zhang et al. [1] study is the most comprehensive recent global measurement study and used a new method based on machine learning to classify uncontaminated observatory stations into rural and urban retaining 12,505 stations out of 33,878 worldwide.

Note that Zhang et al. (2021) observations may likely have been dominated by ISA heating compared to AHR due to the altitude of their observations at about 5–6.5 feet. This suggests possible higher warming issues due to urban areas dominated by cities. As well

Zhang et al. [1] observations were limited to UHI areas. Rural assessments in this study are significant, averaging 38% of the GW ISA influence.

*3.3. Mitigating Urbanization Heat Fluxes*

3.3.1. Natural Reflectivity of Land as UHI Comparison

Satellite measurements assess the impact of urbanization often making an albedo comparison to nearby rural areas which have either modest or no urbanization activities. As we point out, this is a challenging comparison since cities amplify heat making it difficult to equate it to the albedos of rural areas. One might also ask is this the best comparison when judging GW effects? In this study, the reference comparison was to consider ISA temperatures above the earth's average ambient temperature. For Satellite albedo assessments, a likely equivalent alternate comparison would be to consider the global natural average reflectivity of land or earth. He et al. [20] found that the natural land albedo varies from 0.1 to 0.4 with an average of 0.25. In Guo et al. [19] study that found substantial negative albedo-induced radiative forcing in 11 cities in China from 1980 to 2015 averaging $-7.76$ Wm$^{-2}$, only one of the 11 cities studied had an average albedo close to 0.25. The other ten cities averaged about 0.15. Therefore, if comparisons were made relative to the natural land reflectivity, even without amplification taken into account, a large positive albedo radiative forcing would result.

Feinberg ([50] defined albedo heat pollution (i.e., excess LSCT heat flux) in terms of the planet's natural reflectivity of land as any manmade surface or combination of surfaces with resulting lower reflectivity than 25% and with local amplification $\geq 1$. That is, one goal might be to match the natural reflectivity of land to reduce 'heat pollution' (i.e., improper land-cover and land-use) and UHI contributions to global warming. For example, Feinberg [50] showed that if we compare a black asphalt road to the natural average reflectivity of land (25%) with global secondary effect (no local amplification factors), one acre of black asphalt roads equates to 74,200 gallons of gasoline energy per year per acre or about 7.5 times more energy in heat pollution that a solar power plant produces per acre.

It is easy to assume that the neighboring rural albedo areas should be used in GW comparisons. However, this provides a lack of information as there is a serious energy imbalance in urbanization as indicated by UHI $\Delta T$ observations. This is partly the result of local amplified heat flux. The need for local and regional mitigation is apparent from Tables 4–7, Equation (14), and Zhang et al.'s [1] work. Reducing heat fluxes also reduces the potential for local and global amplification.

3.3.2. Estimating the UHI Solar Geoengineering Requirements

We see from Figure 2, the slope is 2196 TWh per year with similar recent increases in the last 5 to 10 years. Therefore, it is difficult to mitigate GW by reducing energy consumption. Improvements in energy efficiency and population development supported by multi-actor governance would be helpful in this area.

In terms of solar geoengineering, the likely candidate is a need to increase the reflectivity of cities. Yet due to dark rooftops and pavements, this is becoming increasingly impossible [63]. A recent MIT [68] pavement study concluded that in all U.S. urban areas, an increased temperature of 1.3 °C occurs in summer months and heatwaves are 41% more intense with 50% more heatwave days due to asphalt pavements. Yet policymakers have not added solar geoengineering albedo guidance in the Paris Agreement. However, based on the work by Zhang et al. [1] and the findings in this study, we see that such goals are urgently needed.

Studies have suggested [69] pavement UHI problems even back in 2005 [70] or earlier and there is a history of UHI authors' findings warning of likely UHI contribution to global warming (see Feinberg, [25] for a review) with no mitigating effort issued in the Paris Agreement. We can estimate feasible UHI-required goals. Although each city can differ in requirements, we can look to assess the general capability of UHIs.

Solar geoengineering of UHIs has been studied and increasing the UHI albedo by about 0.1 is feasible. In a paper by Akbari et al. [69] it was estimated that "white rooftops and light-colored pavements, can increase the albedo of urban areas by about 0.1 . . . . and could potentially offset some of the anticipated temperature increase caused by global warming".

Therefore, perhaps the most practical UHI mitigating goal might be to try and accomplish an albedo increase of 0.1 as indicated by Akbari et al. [69]. UHI albedos are typically in the range of 0.1 to 0.15 [19,65]. Therefore, a 0.1 increase is also close to the 0.25 natural reflectivity of the land. We can assess cooling capabilities relative to this goal for $\Delta T$ that should result. In Appendix B, Equation (A9), we find the applicable relationship for this assessment is

$$X_c \frac{S_o}{4}(\Delta \alpha) = \frac{\varepsilon_{eff}\sigma}{\overline{A}_{UHI\_mixed}}(\overline{T}_S^4 - (\overline{T}_S + \Delta T)^4) \tag{28}$$

Here $\overline{T}$ is the average rural neighboring climate temperature, $\Delta T$ is the average observed LSCT UHI change for an increased albedo for the $\Delta \alpha = 0.1$ goal, $X_c$ is the average irradiance, and $\overline{A}_{UHI\_mixed}$ is from Equation (11). For example, if the average neighboring rural climate temperature is 14.5 °C, and considering an average global irradiance of 0.47 [24,53] and $\overline{A}_{UHI\_mixed} = 3.2$, we can solve Equation (28) finding an LSCT UHI cooling capability for unshaded areas of

$$\Delta T_{LSCT} = -9 \text{ °C } for \ \overline{T}_S = 14.5 \text{ °C, } \Delta \alpha = 0.1 \tag{29}$$

Although this is an LSCT change it relates to the UHI $\Delta T$. Increasing an albedo of 0.1 globally in cities would likely mitigate a large portion of the UHI $\Delta T$ effect and its local and global amplification effects worldwide. Note that without considering the amplification reduction $\overline{A}_{UHI\_mixed}$, $\Delta T_{LSCT} = -2.9$ °C.

From the likely best estimate, road contributions are 1.1% of global warming but may be higher due to a lack of data [28]. If asphalt was replaced with concrete, (with a 5 times more reflective type surface), the global warming cooling would be about a factor of 5 [70] yielding a reduction of at least −5.5% in global warming (5 × 1.1%). Solar geoengineering is challenging, but cool roads and roofs are feasible. An EPA ([71] and a recent MIT [67] study found that cool roads are highly cost-effective when implemented during maintenance and construction periods.

## 4. Discussion

### 4.1. Mciroclimate Amplification Control for City Cooling Requirments

Controlling the microclimate to cool off cities is important. Although the temperature is a good gauge of heat in cities, measuring the strength of the UHF microclimate amplification is likely best measured using the city's footprint. We noted the footprint measures the combined effects of the UHF microclimate amplification effects as illustrated in Figure 1. Zhou et al. [33] found an average footprint of 3.2 times the urban area and warned that "ignoring the footprint may underestimate the UHI intensity in most cases". Yang et al. [39] in a temporal study in China showed that footprints rapidly grow at an alarming rate of about 4.4% per year, indicating the rapid spreading of 'heat pollution' and lack of microclimate control in China's cities. We presented the strength of the UHF microclimate amplification in humid environments with the help of Zhao et al. [16] study that found an average $\Delta T$ increase of 3.3 K observed in daytime hours in humid climates. This data was key in supporting Models 2, 3, and 4. We found that water vapor feedback in cities was about a factor of 2.1 times higher than the standard atmosphere (Equation (9), Model 2). We noted the importance of a study by Liu et al. [23] which found during the COVID lockdown in China cities when traffic was minimal and energy use was low, that surface UHI intensity was reduced by 25%. about 40% reduced in the canopy. This study supports the $CO_2$ Model 1 effect and the strength of AHR in the canopy. These strong local effects add up in cities worldwide and this study found likely significant global warming contributions in Section 3.2.2 of about 13% in agreement with Zhang et al.'s [1] measurements. These key

observations show that city planners should not ignore UHF microclimate amplification effects and respect the need for cool roads and roofs as well as the need for energy efficiency to control AHR and reduce UHF in city microclimates. We find that UHF microclimate control should be an important added part of the Paris Agreement.

*4.2. Satellite Assessments*

Recent satellite-based studies are finding that many cities have enough of a reflective quality that one could conclude the UHI effect may be dominated by AHR compared to the albedo effect and other issues [19,21] using satellite methods found negligible urbanization albedo forcing indicating, "Relative to 2001, the increased urban lands in 2018 produced a radiative forcing of ~0.00017 $Wm^{-2}$". However, Feinberg [25] found that increased albedo changes from urban land from 1950 to 2019 did produce higher radiative forcing using the MCAE of the order found in Equation (21). As well, as assessments in Section 3.2 and Zhang et al.'s [1] study conflict with Ouyang et al.'s [21] results.

Yang et al. [14] found that the annual average anthropogenic heat flux in mainland China provinces averaged 0.4 W/m² and could be as high as 30–40 W/m² on a city scale. Guo et al. [19] found substantial negative albedo-induced radiative forcing from 11 cities in China from 1980 to 2015 averaging $-7.76\ Wm^{-2}$. That is, they observed an albedo-induced UHI cooling effect. We note this means a substantial AHR effect greater than 7.76 $Wm^{-2}$ would be needed to create the observed UHI $\Delta T$ effect in these cities indicating that AHR must be the root cause that dominates city warming trends in the Guo et al. [19] study. For example, one of the cities reported by Guo et al. [19], Shanghai, is close to the average with a negative albedo RF of $-7.26\ Wm^{-2}$ and has a UHI $\Delta T$ rise of at least 1.5–2 °C with an albedo average of about 0.16. This would require AHR forcing roughly of about 15–20 $Wm^{-2}$ (see Table 4 for example). There are many potential satellite issues that may cause difficulties in such measurements discussed in Appendix G.

## 5. Conclusions

We note there are reasonable uncertainties in assigning attributions in global warming estimates as climate modeling is not a perfect science. However, in this study physics-based modeling agreed with data-aided observation for UHF scenarios, as demonstrated in Section 3.1.9, Section 3.1.10, and Section 3.1.10.1. Overall, the results found:

- AHR from 1950–2021 due to energy consumption is estimated to have a maximum GW influence of 6.5% (Equation (14)). This is mainly related to population growth. Here, we see that the energy consumption increase in 2021 was 1.2% and this is highly correlated to a population growth rate [3] that decreased to 0.89% (Section 3.1.8).
- Unshaded ISAs from 1950 to 2019 are estimated to have an average GW influence of 6.5% (Table 7). This is broken down with average contributions of 4.0% from urban ISAs and 2.5% from rural ISAs.
- Heat fluxes from unshaded ISAs and AHR combined are estimated to have an average GW influence of 13% (Table 7) over the approximate time period of about 70 years from 1950.
- The main microclimate amplification factor justified with data was 3.2 (Table 7) in Model 4 (Section 3.1.10). It is assumed that UHIs dominate urban effects.
- Unshaded ISAs that helped match ground-based observations indicated that urban ISA temperatures would likely average 10 °C above a global ambient temperature with an average albedo of 0.133, while rural ISAs were estimated at 11 °C above ambient with an average albedo of 0.115. (Rural ISAs are anticipated to have a higher temperature due to increases in the percentage of asphalt roads and roofs, Table 7).
- The ISA average breakdown was 59% rural and 41% urban (Table 7).
- GHGs with water vapor feedback were found in modeling to be a major amplifier of AHR microclimate heat fluxes increasing UHI $\Delta T$s by about 48% (Equation (19)).
- Roads are estimated to contribute 1.1% to GW but may be higher due to a lack of data and satellite resolution (see Section 3.2.2). New roads were observed to be darker and

smoother and will likely clean better in the rain, therefore, unfortunately, will likely be much hotter over their lifetime maintaining low albedos compared to old roads. The overuse of black asphalt on roads and roofs are highly dangerous to our environment, contributing significantly to urban heat wave intensity, city temperatures, and global warming, suggesting that such practices should be banned.

- Changing roads from asphalt to concrete or similar type surface reflectance can increase their reflectivity by about a factor of 5 and reduce global warming by at least 5.5%.
- Without considering any secondary amplification effects, results indicated that AHR and solar heating of ISAs heat fluxes would equate to about 0.7% and 1% GW influence, respectively.
- A heat flux likely scenario found AHR and unshaded ISAs in cities may average 6.5 $Wm^{-2}$ and 4.0 $Wm^{-2}$ (Equation (20)) respectively totaling a 10.5 $Wm^{-2}$ baseline value and this was estimated to increase UHI $\Delta T$ to about 1.75 °C which could be further amplified in dry and wet microclimates to about 3 °C to 6.3 °C (Table 5), respectively.
- Given average climate conditions, it is possible to mitigate much of the UHI effect with an albedo increase of 0.1 which is anticipated to lower the average impermeable surface temperatures by about 9 °C (Equation (29)) that studies show can be accomplished with cost-effective cool roads and roofs.
- Not accounting for UHFs and their microclimate and global amplification effects may result in climate model attribution errors of 2XUHF influence (1 × UHF due to not including the urbanization influence, and 1 × UHF in overestimating the current GHG influence, as illustrated in Section 3.3). The suggested correction in Appendix F is an urbanization forcing of 0.31 $Wm^{-2}$ and with feedback influence (×2.15) yields a value of 0.66 $Wm^{-2}$. This results in a possible 13% urbanization warming effect that occurred between 1950–2019.
- The forcing estimate for UHF 0.31 $Wm^{-2}$ (Section 3.3) led to an extra increase in atmospheric water vapor averaging 204 ppm (Appendix F, Equation (13))

Results help explain feasible root causes and support Zhang et al.'s (2021) findings. Assessments were made with conservative microclimate Models 1–5 that were aided by data to help justify local climate amplification factors. There is relatively high confidence in background global amplification factors. In modeling, many conservative microclimate values were used primarily in Equations (8) (Model 3), (10)–(12), (21), and (22), so warming estimates may be higher.

We discussed the importance of mitigating the UHF effects in cities worldwide primarily by increasing city albedo to at least the natural reflectivity of land which is about 25% as reported by He et al. [20]. Other authors have suggested that a 0.1 albedo increase is feasible in UHI [69] and can be cost-effective for cool roofs and roads [67].

As well from Figure 2, it is important to reduce energy consumption which appears to have a market increase slope after 1950 showing a yearly increase in consumption of 2196 TWh per year from 1975 to 2021, and this has not slowed significantly in the last 5 to 10 years as energy use is highly correlated with population growth rates [3]. Therefore, it is difficult to mitigate this by reducing energy consumption. Improvements in energy efficiency and population development supported by multi-actor governance would be helpful in this area.

At the current rate of energy consumption, one current estimate indicates we may run out of fossil fuel by 2060 (Octopus Energy [72]).

Zhang et al. [1] and Feinberg [17,25,53] recent findings and results here are unable to support the IPCC in AR6 (2021) statement that urbanization has a "negligible impact on global annual mean surface-air warming (very high confidence)." Furthermore, this is also contrary to a long history of UHI findings by other authors (see Feinberg [25] for an overview). Strong support by the IPCC in recognizing the urbanization heat flux GW risks occurring is lacking and is badly needed as it is likely an important influence on policymakers. Policymakers for many years have not implemented any worldwide Paris Agreement albedo goals. Without IPCC support to help recognize this issue the problem of

poor solar urbanization color choices and yearly increases in black asphalt roads, dark roofs, and building sides will continue which has already made the task of solar geoengineering of urbanization nearly impossible [53]. Currently, negative solar geoengineering dominates urbanization worldwide. Results also point to the conclusion that the overuse of black asphalt on roads and roofs are highly dangerous to our environment contributing to city heatwaves and higher temperatures (Sections 3.3.1 and 3.3.2), and global warming findings suggest that such practices should be banned. The author noted that new roads are smoother and clean better in the rain and will likely maintain higher temperatures over their lifetime.

The urbanization heat flux concern should be at least at the level of $CO_2$ mitigation, especially with city growth rates. It is equally important as a helpful solution for many reasons including the albedo mitigating advantage [9,53], its ability to quickly reduce local and partial global temperatures, and the difficulty of mitigating $CO_2$. As well, energy consumption increases will make GW mitigation difficult. We suggest that the Paris Agreement albedo goals detailed in Section 4.4 and discussed in studies such as Feinberg [53], are urgently needed and should be provided as soon as possible.

**Funding:** This research received no external funding.

**Data Availability Statement:** All data is provided within this paper. There are no external sources other than the references listed.

**Acknowledgments:** The author would like to thank the journal Land's open access stipend for this work.

**Conflicts of Interest:** The author declares no conflict of interest.

**Abbreviations**

anthropogenic heat release (AHR), footprint (FP), global warming (GW), greenhouse gases (GHGs), heatwave (HW), impermeable surface areas (ISAs), land surface contact temperatures (LSCTs), loss of cooling (LC), methods of climate amplification estimates (MCAE), solar geoengineering (SG), urbanization heat fluxes (UHF), urban heat island (UHI), upwelling (UW), water vapor feedback (WVF).

**Appendix A.**

The applicable equation used for Equations (17), (18), and (24), and in Tables 4–7 results are

$$\Delta \overline{P}_{Forcing} = \left( \overline{P}_U \right) - \left( \overline{P}_{Ru} \right) = \varepsilon_{eff}\sigma\left( \left( \overline{T}_{Ambient} + \Delta T_U \right)^4 - \left( \overline{T} \right)^4_{Ambient} \right) \tag{A1}$$

Here the subscripts U refers to urban and Ru to rural. Solving for $\Delta T_U$ we obtain

$$\Delta T_U = \left[ \frac{\left( \Delta \overline{P}_{AHF\_UHI} + \sigma\left( \overline{T} \right)^4_{Ambient} \right)}{\varepsilon_{eff}\sigma} \right]^{1/4} - \overline{T}_{Ambient} \tag{A2}$$

where $\Delta \overline{P}_{AHF\_UHI} = \Delta \overline{P}_{Forcing}$

**Appendix B. Albedo UHI Mitigation Assessment**

For a given albedo $\alpha_1$ associated with a temperature $T_1$, their relationship can be expressed by

$$\frac{S_o}{4}(1 - \alpha_1) = \varepsilon_{eff}\sigma T_{1S}^4 \tag{A3}$$

Given an albedo change to $\alpha_2$ associated with a temperature change to $T_2$ we can write

$$\frac{S_o}{4}(1 - \alpha_2) = \varepsilon_{eff}\sigma T_{2S}^4 \tag{A4}$$

The difference from Equations (A3) and (A4) yields

$$\frac{S_o}{4}(\alpha_2 - \alpha_1) = \sigma(T_{1S}^4 - T_{2S}^4) = \varepsilon_{eff}\sigma(T_{1S}^4 - (T_{1S} + \Delta T)^4) \tag{A5}$$

where

$$T_{2S} = T_{1S} + \Delta T \tag{A6}$$

Equation (A5) provides a relationship between temperature and albedo changes relative to the natural reflectivity of the Earth without clouds.

$$\frac{S_o}{4}(\Delta \alpha) = \varepsilon_{eff}\sigma(\overline{T}_S^4 - (\overline{T}_S + \Delta T)^4) \tag{A7}$$

where

$$\Delta \alpha = \alpha_2 - \alpha_1 \tag{A8}$$

Because the LHS = RHS in Equation (A7), we are free to study either side in terms of global warming changes. Consider now an ISA denoted as $A_{ISA}$, having an irradiance factor $X_c$ and with local amplification factor A, we write the relation for the UHI mitigating albedo to its $\Delta T$ observation as

$$\Delta \overline{p}_{Forcing}(\text{watts}) = X_C A A_{ISA}\sigma\varepsilon_{eff}(\overline{T}_S^4 - (\overline{T}_S + \Delta T)^4) \tag{A9}$$

These results are used in Equations (24)–(27). We can divide B7 by the area of the earth to obtain the global warming influence. Equation (A9) was originally developed by the author and is an alternate form of Equation (12) in the reference by Feinberg [53].

## Appendix C. Local vs. Global Water Vapor Feedback Amplification Estimate Comparisons

*Appendix C.1. Global Water Vapor Feedback Amplification*

Gordon et al. [30] estimated a water vapor feedback of 2.4 $\text{Wm}^{-2}\,\text{K}^{-1}$ at a global temperature of nearly 14 °C. For a 1 K global temperature rise, this would yield an amplification of

$$A_{Water\_Vapor} = \frac{5.1\ \text{Wm}^{-2}}{5.1\ \text{Wm}^{-2} - 2.4\ \text{Wm}^{-2}\text{K}^{-1}\ \times\ 1.0\ \text{K}} = 1.81 \tag{A10}$$

Liu et al. [23] estimated a water vapor feedback of 1.6 $\text{Wm}^{-2}\,\text{K}^{-1}$ at a global temperature near 14 °C. For a 1 K global temperature rise, using Liu et al.'s [23] estimate in Equation (A10), this would yield an amplification of 1.5. In this paper, the full feedback is assessed at 2.15 [9,17] which includes WVF and other issues.

*Appendix C.2. Local Water Vapor Feedback Amplification*

Feinberg [17] estimated a water vapor feedback of 3.4 $\text{Wm}^{-2}\,\text{K}^{-1}$ at a UHI temperature of 15 °C. For a 1 K UHI temperature rise, with a comparable local heat flux used in Equation (A10), results are

$$A_{Water\_Vapor} = \frac{5.1\ \text{Wm}^{-2}}{5.1\ \text{Wm}^{-2} - 3.4\ \text{Wm}^{-2}\text{K}^{-1}\ \times\ 1.0\ \text{K}} = 3 \tag{A11}$$

This illustrates that the local water vapor feedback is about a factor of 2 higher than the global average. Most UHIs have a $\Delta T$ increase higher than a 1 K rise. This would increase the local amplification effect.

**Appendix D. UHI CO$_2$ Surface & Dome Re-Radiation**

Koerner and Klopatek [7] estimated about 80% of CO$_2$ originates in urban environments which could be a strong local re-radiation amplification effect. Re-radiation from CO$_2$-IR interactions at the surface and in the canopy microclimate is not well understood. However, the likelihood will be primarily correlated to IR 667 cm$^{-1}$ emissions and CO$_2$ concentration levels. CO$_2$ concentrations in the UHI canopy are generally much higher than in rural areas. As well, many authors studying CO$_2$ concentration find a high correlation to vehicle traffic emissions. The idea that CO$_2$ re-radiation of surface IR and AHR in the canopy urban microclimate can create amplification simply extends greenhouse gas theory.

Studies show there is a high CO$_2$ concentration near the surface and in the canopy areas compared to rural areas. In France, Widory and Javoy [73] measured CO$_2$ concentrations in Paris, its suburbs, and the surrounding open countryside, as well as from vehicles, heating sources, power stations, etc. outside Paris that averaged 415 ppm, while values in the city sometimes reached as high as 950 ppm (229% increase). These higher values were driven primarily by vehicle exhaust. In Italy, Gratani and Varone [74] measured atmospheric CO$_2$ concentrations at various places throughout Rome in 1995, 1998, and 2001–2004. The data they obtained pointed to the existence of a higher amount of variable CO$_2$ in the UHI dome area. CO$_2$ concentrations at the periphery of the city averaged 405 ppm and 377 ppm on weekdays and weekends, respectively, whereas at the city center, they averaged 505 ppm and 414 ppm on weekdays and weekends, respectively (about a 25% increase). In addition, there was a strong correlation between traffic density and CO$_2$ concentration.

George et al. [75] in Baltimore measured differences in the city's center (urban site), a suburban site, and a rural site over 5 years. Atmospheric CO$_2$ was significantly increased on average by 66 ppm (about 19% higher) from the rural to the urban site and air temperature was also significantly higher at the urban site (14.8 °C) compared to the suburban (13.6 °C) and rural (12.7 °C) sites. Results were related to population and associated high traffic volume.

In Mexico, Velasco et al. [76] studied a densely populated area and CO$_2$'s highest concentration average of 421 ppm and the lowest concentrations average of 375 ppm (12%). Velasco et al. [73] found that CO$_2$ concentrations were "directly related to vehicular traffic, vehicles accounted for approximately 60% of the urban CO$_2$ emissions.

Liu et al. (2022) found "a substantial decline in both surface and canopy UHIs over 300-plus megacities in China during lockdown periods compared with reference periods". During lockdown measures, "the surface UHII was reduced by 0.25 ± 0.22 K in the daytime and 0.23 ± 0.20 K at night. The reductions in canopy UHII during lockdown periods are more significant, reaching 0.42 ± 0.26 K in the daytime and 0.39 ± 0.29 at night. The linkage between lockdown measures and UHI variations provides new insight into the influence of this pandemic on urban climate." They further state, "It is therefore convincing that the observed UHII decreases during lockdown periods are mainly associated with reduced AHR induced by the closure of road transportation, shutdown of nonessential business and industrial activities, and restrictions of outdoor activities".

Feinberg [50] in a molecular-level study on UHI microclimates indicated the capability of CO$_2$ to amplify heat flux in UHIs can be significant in the urban climate and can impact warming at the global level. UHI dome-surface microclimate areas where CO$_2$ active pollution occurs at much higher levels combined with intense IR radiation were assessed. Estimates only required about 780 times more frequent CO$_2$-IR interactions per molecule per year to achieve IPCC forcing in a 'what if' scenario. Estimates were made by considering a 16% increase in CO$_2$ compared to the standard atmosphere. The UHI volume assessed had a fractional area of 0.188 of the earth with a height of 100 m. Assessment appeared to indicate that only the CO$_2$ concentration may not be the limiting factor. It was more likely that available heat flux from anthropogenic heat release and albedo absorption in UHIs limits the amount of local CO$_2$ re-radiation that can intensify microclimate warming in the UHI and its related effect on global warming.

### Appendix E. Model 3 Humidity Correction for Dry Areas

The humidity correction for the dry region estimate of 33%, actually consists of 11% sub-humid and 15% semiarid; these conditions are detailed in the equation below.

$$\overline{A}_{UHI\_mixed} = 0.11A_{U-Subhumid} + 0.15A_{U-SemiArid} + 0.07A_{U-Arid} + 0.67A_{U-Wet}$$
$$A_{U-Subhumid} = 2.125 \times 0.83 = 2.54, A_{U-SemiArid} = 2.125 \times 0.5 = 1.28, A_{U-Arid} = 1$$
$$\overline{A}_{UHI\_mixed} = (0.11(1.76) + 0.15(1.0625) + 0.06(1) + 0.67(2.125))1.2 \qquad \text{(A12)}$$
$$= (0.1936 + 0.1594 + 0.07 + 1.424)1.2 = 2.2$$

The weighting factors for subhumid are due to NOAA climate ratios [77]. The chart provided in this reference finds an average $P/E_p$ ratio of 1.2 for humid, 1 for subhumid, 0.6 for semiarid and we use arid for dry areas. Here P is precipitation (in mm) and $E_p$ is the potential evapotranspiration (in mm). We then normalize this for humid areas yielding 1 for humid, 0.83 for subhumid, and 0.5 for semiarid. These are shown in Equation (A12). Note the 2.125 factor is from Model 2 and the 1.2 factor is from Model 1 Equation (8). Results show a 5% correction yielding a small increase from 2.1 to 2.2 overall in Equation (11).

### Appendix F. Water Vapor Feedback and Radiation Energy Flux Breakdown with UHF

The consequences of a 13% increase in global warming due to urbanization introduces key issues both in making radiative warming estimates and how it affects atmospheric water vapor. When the global air temperature increases, it causes an expansion of atmospheric gases and allows the air to hold more water vapor before saturation. Thus, warmer air holds more water vapor. For example, from 1950 to 2019, a 0.95 °C rise occurred. The rule of thumb is the increase in water vapor goes as the temperature ratio, in this case

$$T_2/T_1 = 15.45\,°\text{C}\,/14.5\,°\text{C} = 1.0655 \qquad \text{(A13)}$$

Here $T_1$ = 14.5 °C, the 1950 estimated average air temperature, and $T_2$ = 15.45 °C, the increase in 2021. We can refine this with the Clausius-Clapeyron relative humidity relationship

$$Water\ Vapor\ Factor = Exp[-2.465\,E\,6/462\{1/T_1 - 1/T_2\}] = 1.0629 \qquad \text{(A14)}$$

Here $T_1$ and $T_2$ are in degrees K, 2.465E6 J-kg$^{-1}$ is the latent heat of vaporization and 406 J-kg$^{-1}$ K$^{-1}$ (287) is the specific gas constant for water vapor. We note Equations (A13) and (A14) are in good agreement.

The average estimated composition of the atmosphere up to an altitude of 25 km with $CO_2$ is about 400 ppm and water vapor is about 25,000 ppm [78].

Using this water vapor value, and Equation (A14), the average will increase by 1573 ppm (=25,000 × 1.0629–25,000) since the average air is warmer by the 0.95 °C temperature rise.

Then given our results that the UHF forcing is about 13%, this portion yields 204 ppm, leaving 1368 ppm due to the general warming of GHGs leaving a breakdown for the increase in average water vapor increase as

$$Water\ Vapor\ Increase\ (1950\ to\ 2021) = (204\,\text{ppm})_{UHF} + (1368\,\text{ppm})_{GHGs} \qquad \text{(A15)}$$

We have been using the radiation warming increase from 1950 to 2018 of 5.1 Wm$^{-2}$. If we extrapolated from IPCC/NOAA tables for the period between 1950 and 2019, yielding 2.38 W/m$^2$ estimated by Butler and Montzka [79]; see also Feinberg, [9] the estimates for change in the radiation energy flux $\Delta R$, yields the following breakdown

$$\Delta R = \left(1.78\text{Wm}^{-2}\right)_{CO_2} + \left(0.6\text{Wm}^{-2}\right)_{\text{other GHGs}} + \left(2.72\text{Wm}^{-2}\right)_{H_2O} = 5.1\text{Wm}^{-2} \quad \text{(A16)}$$

The problem is, this IPCC estimate does not account for the UHF energy contribution that is estimated here as 0.66 Wm$^{-2}$ (13% of 5.1 Wm$^{-2}$). This includes its feedback effect.

Without feedback it forcing is 0.307 Wm$^{-2}$. This is the UHF forcing (=0.66 Wm$^{-2}$/2.15). We can consider the following proportionate correction reducing other by GHGs 25% and $CO_2$ by 75% of the UHI value. Then Equation (A16) changes to

$$\Delta RW = \left( \begin{array}{c} \left(1.55\text{Wm}^{-2}\right)_{CO_2} + \left(0.52\text{Wm}^{-2}\right)_{\text{other GHGs}} \\ + \left(0.31\text{Wm}^{-2}\right)_{\text{UHF}} \end{array} \right)_{Forcing} + \left(2.72\text{Wm}^{-2}\right)_{\text{Feedbk}} \tag{A17}$$
$$= 5.1\text{Wm}^{-2}$$

Note that the 0.31 Wm$^{-2}$ urbanization forcing represents 13% of the forcing (=0.31 Wm$^{-2}$/(1.55 + 0.52 + 0.31) Wm$^{-2}$). The attribution change is 13% less GHG and 13% added urbanization effect. Note this 13% ratio is maintained just as the full urbanization warming contribution with a feedback value of 0.66 Wm$^{-2}$ (=2.15 × 0.31 Wm$^{-2}$) is also 13% of total warming feedback and forcing 5.1 Wm$^{-2}$ flux.

We can assess our carbon dioxide correction estimate using a forcing relationship by Myhre et al. [80] (see also ASC [81]) with values of 311 ppm in 1950 and 415 ppm in 2019 for $CO_2$. This indicates a radiation-forcing estimate of

$$1.54\text{Wm}^{-2} = 5.35\text{Wm}^{-2} \text{ Ln}(415/311) \tag{A18}$$

This is fairly close to our estimate for $CO_2$ forcing correction estimate of 1.55 Wm$^{-2}$ in Equation (A17), indicating some confidence in our estimate.

We can also use Equation (A18) for water vapor if we know its relative strengths compared to $CO_2$. Kiehl et al. [82] estimated that water vapor was about 2.3 times stronger than $CO_2$ forcing on a clear sky day, but larger on cloudy days (about 33% larger). Consider the estimated increase in the average water vapor of 25,000 ppm of 1573 ppm in Equation (A15). This average radiation warming could be compared to an area having 20,000 ppm of water vapor. We can compare these two areas with different water vapor IR downwellings by using Equation (A18) in a similar way by applying a $CO_2$ relative strength factor of f = 3 to water vapor yielding a very rough estimate of,

$$5.35 \text{ Wm}^{-2}\{(\text{Ln}[(25,000 + 1573 \times \text{f})/20,000)]\}) = 2.1 \text{ Wm}^{-2} \tag{A19}$$

This is just a helpful example demonstrating the logarithmic saturation of water vapor that provides an idea of how a substantial increase of 1573 ppm compared to a 100 ppm increase of $CO_2$, only leads to a radiative warming effect of 1.55 Wm$^{-2}$ estimate in Equation (A17). This suggests a possible average scenario. However, the purpose of the example is to help illustrate possible warming causality and also the difficulty of climate science in making such estimates.

### Appendix G. Satellite Issues in Urbanization Assessments

Satellite measurements such as those reported by Guo et al., [19] and Ouyang et al. [21] conclude a negligible or even a reverse forcing albedo effect from urbanization. However, Ouyang et al. [21] quoted the area assessed as about 0.15% of the earth which also include farmlands. This is about 1.7 times smaller than Sun et al.'s [27] estimate for ISAs. This is likely partly due to the difference in accounting for ISA building sides. Ouyang et al. [21] suggest that effective radiative forcing (ERF) modeling may be a better tool than radiative forcing (RF) assessment, but provide assessments with simple RF measurements that ignore complex vertical building structure sides with solar heating of ISAs in UHIs. Satellite results make comparisons to rural areas. However, heat fluxes in cities are subjected to amplification effects and these need to be considered. Furthermore, in this study, LSCT assessments were an added metric that is part of our everyday experiences of solar heating of ISAs heat fluxes in city pavements, roofs, building sides, and on millions of miles of rural black asphalt roads. This was assessed relative to ambient conditions. In this study, we focus on ISAs, not all urbanization albedos which can include farmland for example.

To this point, Kiersten and Möller [18] note that "Remote sensing has several limitations including capturing emissions from vertical surfaces" such as the sides of building walls. This is a challenge for remote sensing because the equipment is designed to capture missions from horizontal surfaces such as streets, rooftops, and treetops [83]. The sensor captures data from a birds-eye point of view which is not always the most accurate representation of the UHI. The second challenge is that remotely sensed data represents radiation that has traveled through the atmosphere twice [83]. As the sensor receives the measurement, the wavelength has traveled from the sun to the earth and then reflected from the earth's surface back into the atmosphere to an orbiting or high flight sensor. To account for this error the data must be adjusted to include solar reflectance and temperature [83]. He et al. [20] found large albedo satellite discrepancies: "At latitudes, higher than 50°, the maximal difference in winter zonal albedo ranges from 0.1 to 0.4 among the nine satellite data sets". A study in optimum conditions with clear sky and reasonably flat terrain of the Atacama Desert by Cordero et al. [84] found: "Satellite-derived estimates for sites in the southernmost part of the Atacama Desert (27–30° S) exhibited negative relative bias errors (up to about −20%). These larger discrepancies can be attributed to the scattered vegetation in the region and the difference in MODIS and our instruments . . . The relatively lower signal-to-noise ratio of our ground measurements . . . the reflectance (and in turn the signal) was relatively low over the range 2105–2155 nm due to a water vapor absorption band . . . " These types of numerous challenges in Ouyang et al. [21] satellite approach are not well addressed in their paper. By comparison, Zhang et al. [1] provided a detailed history of ground-based measurement issues and their precautions.

The satellite measurements may not accurately assess hot spot areas that likely dominate warming issues because they raise the ISA average temperature value and may require higher resolution to estimate actual global averages. As mentioned a solar-heated rural road and one in a city can almost be equal in temperature within the ambient temperature differences which in comparative assessments values may be compromised. As well, assessing albedo can be difficult; ground measurements find very large albedo pavement variations seasonally, due to aging, and material type. Therefore, satellite results may have large variabilities that are not addressed.

Theobald et al. [56] argued that "remotely sensed imagery has limitations because it can require human interpretation to classify adequately and can miss development features that are obstructed by vegetation canopy or are small or narrow features".

Satellite results assessed by Ouyang et al. [56] did not recognize the results in the Zhang et al. [1] study. Furthermore, a full assessment of contact surface temperature observations worldwide may need very high resolutions to resolve all asphalt streets and roof temperatures. As well building sides are likely a key issue where satellite measurements are unable to accurately assess. Furthermore, UHIs amplify heat from the albedo effect which should be taken into account when making comparisons to neighboring rural areas. For comparisons, we discuss the relative natural reflectivity of land (~25%) in Section 3.3.1 and how this may be insightful. These are just some of the issues that may be problematic in remote sensing assessment and could be key reasons for their conflicts with Zhang et al.'s results and Table 7. We discuss this as many authors (IPCC) favor satellite albedo land cover assessments ignoring direct measurement ground-based studies that intrinsically include amplification effects, and it is apparent these create key discrepancies.

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
