# Peer review of "Urbanization Heat Flux Modeling Confirms It Is a Likely Cause of Significant Global Warming: Urbanization Mitigation Requirements"

_land, doi:10.3390/land12061222_

Round 1
Reviewer 1 Report (Previous Reviewer 1)
The revisions are satisfactory.
Author Response
Thank you for reviewing my manuscript. As there is no attached file with separate requests, I have not provided any further response.
Reviewer 2 Report (Previous Reviewer 2)
The author generally accepted my requests, and edited a paper, which is now much better than before. The paper contains the explanations wherever I missed them. Besides them, the author added several paragraphs to specify some more ideas of the paper.
However, as a result of the author’s efforts, the paper became rather long...

Author Response
Thank you for your peer review. I have attached my responses.

Round 2
Reviewer 2 Report (Previous Reviewer 2)
The Author made the needed corrections. For the flowchart I meant a traditional diagram, but the interactive table using added by the Author is also acceptable. I recommend to publish the paper in its present form.
This manuscript is a resubmission of an earlier submission. The following is a list of the peer review reports and author responses from that submission.
Round 1
Reviewer 1 Report
The paper makes the correct point that the IPCC has downplayed urbanization effects on surface temperature data products, and has not adequately quantified the role of urbanization in climate models. A further and equally valid point by the author is that estimates of the role of urbanization ignore potential amplification processes. It is well known in climate modeling that CO2 itself does not do much warming, instead feedback-driven amplification yields most of the warming. Feedbacks must therefore also be applied to estimates of urbanization effects. The author examines UHI-feedback processes and albedo modification and finds they account for about 12.3% of warming, roughly matching a recent study based on comparison of urban and rural sites.
The author should emphasize more clearly that his physics modeling is very ad hoc and simplified.
There are too many acronyms being used and the discussion is hard to follow in places.
Equation (2) above line 202: the first number should be 176,431
Author Response
Please see attached file for reviewer 1.

Reviewer 2 Report
Please read in the attachment!

Author Response
Please see attached file for reviewer 2.

Reviewer 3 Report
The followings are some of my feedbacks.This is an interesting research arguing the non-negligible impact of urbanization to global warming.
1. This manuscript was prepared in an unusual way as if the author is actually having a conversation with the readers. This is good, yet it contains too many specific references which complicate interpretation.
2. The data you used should be reported in a separate section.
3. Significant rural heterogeneity necessitates caution in rural reference selection. You should report the detailed information of your rural references.
4. You may shorten the Conclusion part to highlight your own key findings.
Author Response
Please see attached file for reviewer 3.
